# The kinesin KIF4 mediates HBV/HDV entry through the regulation of surface NTCP localization and can be targeted by RXR agonists in vitro.

**Sameh A. Gad**[1,2,3], **Masaya Sugiyama**[4], **Masataka Tsuge**[5], **Kosho Wakae**[1], **Kento Fukano**[1], **Mizuki Oshima**[1,6], **Camille Sureau**[7], **Noriyuki Watanabe**[1], **Takanobu Kato**[1], **Asako Murayama**[1], **Yingfang Li**[1], **Ikuo Shoji**[8], **Kunitada Shimotohno**[9], **Kazuaki Chayama**[10,11,12], **Masamichi Muramatsu**[1], **Takaji Wakita**[1]*, **Tomoyoshi Nozaki**[2], **Hussein H. Aly**[1]*

**1** Department of Virology II, National Institute of Infectious Diseases, Tokyo, Japan, **2** Department of Biomedical Chemistry, Graduate School of Medicine, The University of Tokyo, Tokyo, Japan, **3** Department of Microbiology and Immunology, Faculty of Pharmacy, Minia University, Minia, Egypt, **4** Genome Medical Sciences Project, National Center for Global Health and Medicine, Chiba, Japan, **5** Natural Science Center for Basic Research and Development, Hiroshima University, Hiroshima, Japan, **6** Graduate School of Science and Technology, Tokyo University of Science, Noda, Japan, **7** Institut National de la Transfusion Sanguine, Paris, France, **8** Center for Infectious Diseases, Kobe University Graduate School of Medicine, Kobe, Japan, **9** Center for Hepatitis and Immunology, National Center for Global Health and Medicine, Chiba, Japan, **10** Collaborative Research Laboratory of Medical Innovation, Graduate School of Biomedical and Health Sciences, Hiroshima University, Hiroshima, Japan, **11** Research Center for Hepatology and Gastroenterology, Graduate School of Biomedical and Health Sciences, Hiroshima University, Hiroshima, Japan, **12** RIKEN Center for Integrative Medical Sciences, Yokohama, Japan

* wakita@nih.go.jp (TW); ahussein@nih.go.jp (HHA)

**Data Availability Statement:** All relevant data are within the manuscript and its Supporting Information files.

## Abstract

Intracellular transport via microtubule-based dynein and kinesin family motors plays a key role in viral reproduction and transmission. We show here that Kinesin Family Member 4 (KIF4) plays an important role in HBV/HDV infection. We intended to explore host factors impacting the HBV life cycle that can be therapeutically addressed using siRNA library transfection and HBV/NLuc (HBV/NL) reporter virus infection in HepG2-hNTCP cells. KIF4 silencing resulted in a 3-fold reduction in luciferase activity following HBV/NL infection. KIF4 knockdown suppressed both HBV and HDV infection. Transient KIF4 depletion reduced surface and raised intracellular NTCP (HBV/HDV entry receptor) levels, according to both cellular fractionation and immunofluorescence analysis (IF). Overexpression of wild-type KIF4 but not ATPase-null KIF4 mutant regained the surface localization of NTCP and significantly restored HBV permissiveness in these cells. IF revealed KIF4 and NTCP colocalization across microtubule filaments, and a co-immunoprecipitation study revealed that KIF4 interacts with NTCP. KIF4 expression is regulated by FOXM1. Interestingly, we discovered that RXR agonists (Bexarotene, and Alitretinoin) down-regulated KIF4 expression via FOXM1-mediated suppression, resulting in a substantial decrease in HBV-Pre-S1 protein attachment to HepG2-hNTCP cell surface and subsequent HBV infection in both HepG2-hNTCP and primary human hepatocyte (PXB) (Bexarotene, $IC_{50}$ 1.89 ± 0.98 μM) cultures. Overall, our findings show that human KIF4 is a critical regulator of NTCP surface transport and

**Funding:** The study is supported by grants from the Research Program on Hepatitis from the Japan Agency for Medical Research and Development (AMED) (21fk0310104j0905, 21fk0310109j0405 to HHA; 21fk0310103j0305 to TW, and 20fk0310109h0004 to KC) and by grant from the Japan Society for the promotion of Science (Grant in Aid for Scientific Research (19K07586) to HHA); SAG was the recipient of the Egyptian Japanese Education Partnership-3 (EJEP-3) PhD scholarship provided by the Egyptian Cultural affairs and Missions Sector, Higher Ministry of Education, Egypt, URL: https://cdm.edu.eg/cdm/ejep-2/ejep. The funders had no role in study design, data collection and analysis, decision to publish, or preparation of the manuscript.

**Competing interests:** The authors have declared that no competing interests exist.

localization, which is required for NTCP to function as a receptor for HBV/HDV entry. Furthermore, small molecules that suppress or alleviate KIF4 expression would be potential antiviral candidates targeting HBV and HDV entry.

## Author summary

Understanding HBV/HDV entry machinery and the mechanism by which NTCP (HBV/HDV entry receptor) surface expression is regulated is crucial to develop antiviral entry inhibitors. We found that NTCP surface transport is mainly controlled by the motor kinesin KIF4. Surprisingly, KIF4 was negatively regulated by RXR receptors through FOXM1-mediated suppression. This study not only mechanistically correlated the role of RXR receptors in regulating HBV/HDV entry but also suggested a novel approach to develop therapeutic rexinoids for preventing HBV and/or HDV infections in important clinical situations, such as in patients undergoing liver transplantation or those who are at a high risk of HBV infection and unresponsive to HBV vaccination.

## Introduction

Hepatitis B virus (HBV) affects about 250 million individuals globally and is a major cause of chronic liver inflammation. Cirrhosis, liver failure, and liver cancer can all result from a protracted condition of hepatic inflammation and regeneration [1]. Sodium taurocholate cotransporting polypeptide (NTCP) was discovered in 2012 to be a key cellular receptor for HBV and its satellite hepatitis delta virus (HDV), which shares the same envelope as HBV [2,3]. When HBV infects human hepatocytes, its nucleocapsid is carried to the nucleus, where the partially double-stranded rcDNA genome is repaired to covalently closed circular (ccc) DNA. This episomal DNA acts as a template for all viral transcripts and pregenomic RNA, forming a very stable minichromosome that is the primary cause of chronic HBV infection, the generation of antiviral escape mutants, or relapse after ceasing nucleo(t)ide analog anti-HBV treatment [4].

Kinesins are a vast protein superfamily that is responsible for the movement of numerous cargos within cells such as membrane organelles, mRNAs, intermediate filaments, and signaling molecules along microtubules [5]. Kinesins are also thought to regulate cell division, cell motility, spindle assembly, and chromosomal alignment/segregation [6,7]. KIF4 is a highly conserved member of the kinesin family [8–10]. KIF4 is also known to move to the nucleus during mitosis, where it interacts with chromatin to alter spindle length and control cytokinesis [11]. KIF4A has previously been shown to improve the transport of HIV and adenovirus capsids early in infection [12,13]. As a result, KIF4A might be a promising antiviral target. The transcriptional activator Forkhead box M1 (FOXM1) has been shown to increase KIF4A expression in hepatocellular carcinoma (HCC) [14]. Interestingly, HBV upregulates KIF4 expression in HepG2 hepatoma cells, and it was reported to be considerably higher in HBV-associated liver malignancies [15]; no information on the role of KIF4 in HBV infection is currently known.

We performed functional siRNA screening using an HBV reporter virus and HepG2-hNTCP cells to uncover host factors that impact the HBV life cycle. We identified KIF4 as a positive regulator for the early phases of HBV/HDV infection based on the findings of this screen. Further investigation indicated that KIF4 is a critical component in the transport and surface localization of NTCP, where it can function as a receptor for HBV/HDV

entry. RXR agonists like Bexarotene reduced KIF4 expression and HBV/HDV infection by targeting FOXM1. This is the first study to show that KIF4 plays an essential role in HBV/HDV entry and that it may be used to build effective anti-HBV entry inhibitors.

## Results

### KIF4 is a proviral host factor required for the early stages of HBV infection

We previously used HBV particles containing a chimeric HBV genome (HBV/NL), in which the HBV core region is substituted by the NanoLuc (NL) gene, to infect HepG2-hNTCP cells formerly transfected with a druggable genome siRNA library two days before HBV/NL infection. We looked at 2,200 human genes to see if they had any effect on the HBV life cycle [16]. HBV/NL does not replicate because the HBV core protein (HBc) is not expressed, and the NL levels released after infection only represent the early phases of HBV infection, from viral entry to transcription of HBV pregenomic RNA (pgRNA). For each plate, nontargeting or anti-NTCP siRNAs were employed as controls (Fig 1A). The XTT (2,3-bis-[2-methoxy-4-nitro-5-sulfophenyl]-2H-tetrazolium-5-carboxanilide) test was used to assess cell viability; wells with ≥20% loss of cell viability were removed from further investigation. Previously, we discovered host factors with anti-HBV action [16]. In this paper, we describe the discovery of new host factors (proviral factors) necessary in the early stages of HBV infection. The independent silencing of just 14 of the 2,200 host genes (0.6%) reduced NL activity by more than 70% (average of three distinct siRNAs) (Fig 1B). These genes were identified as pro-HBV host factors (NTCP, SAT1, DVL3, AOX1, DGKH, CAMK1D, CHEK2, AK3, ROCK2, RYK, KIF4, TFIP11, SLC39A6, MXRA5). KIF4 was previously shown to be induced in vitro by HBV expression [15]. We also looked at data from an accessible database [17] and discovered that KIF4 was expressed at considerably greater levels in patients with persistent HBV infection than in healthy people ($P < 0.001$). (Fig 1C). The role of KIF4 in HBV life cycle has not yet been reported; therefore, we performed further investigation to clarify it. Silencing KIF4 expression with two distinct siRNA sequences (Fig 1E) led to a 2-fold ($P < 0.001$) or 3-fold ($P < 0.001$) reduction in NL activity relative to cells transfected with the control siRNA (Fig 1D).

### KIF4 is required for cell culture-derived HBV (HBVcc) infection

We used authentic HBV particles generated from grown HepAD38.7-Tet cells to validate the relevance of KIF4 in HBV life cycle (HBVcc). We reduced KIF4 expression in HepG2-hNTCP cells by transfecting particular siRNA 72 hours before HBVcc infection (Fig 2A). At 10–13 days post infection (pi), we examined its influence on hepatitis B surface antigen (HBsAg), cccDNA, DNA, and HBc levels. As a positive control, si-NTCP was utilized. Silencing KIF4 expression resulted in a substantial decrease of both HBsAg in the culture supernatant ($P < 0.001$) and HBV cccDNA ($P < 0.01$) to levels equivalent to silencing NTCP expression (Fig 2B). siKIF4 also decreased total HBV DNA levels as measured by Southern blot analysis (Fig 2C) and inhibited HBc expression as measured by immunofluorescence (IF) (Fig 2D) to levels equivalent to si-NTCP without compromising cellular viability (Fig 2E). We employed primary human hepatocytes (PXB) cultures to examine the effect of suppressing KIF4 expression on HBV infection in a more physiologically relevant paradigm (Fig 2F). As predicted, inhibiting KIF4 or NTCP expression dramatically reduced both HBsAg ($P < 0.01$) and HBeAg ($P < 0.001$) levels (Fig 2G) and decreased both extracellular HBV DNA ($P < 0.001$) and HBV cccDNA ($P < 0.01$) levels, as detected by real-time PCR (Fig 2H). Fig 2I depicts the silencing efficiency of KIF4 siRNA in PXB cells.

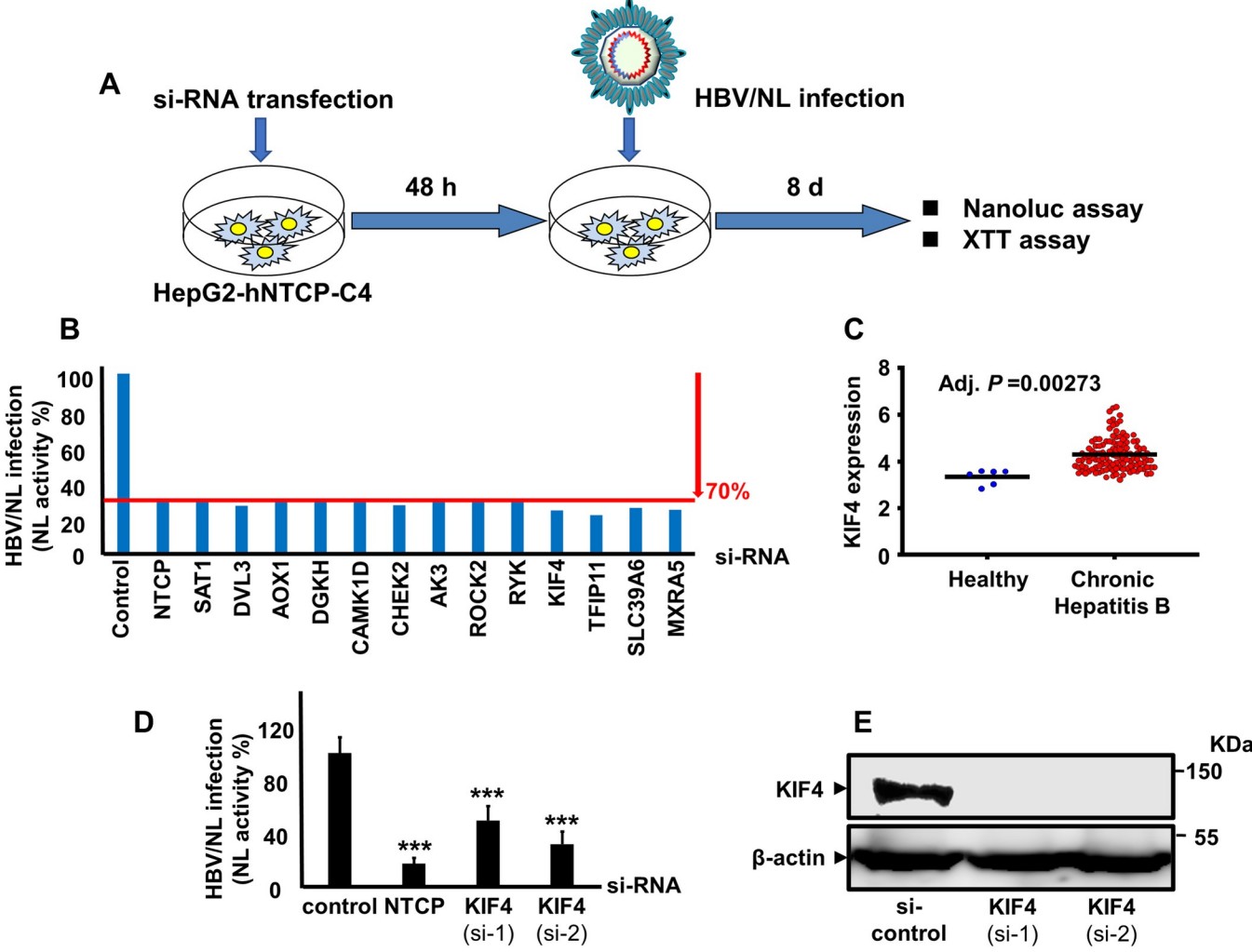

**Fig 1. KIF4 is a proviral host factor required for HBV infection and its expression is enhanced in chronic hepatitis B patients. (A)** A schematic representation of the experimental HBV/NL infection schedule in HepG2-hNTCP used for siRNA library screening. **(B)** HepG2-hNTCP cells were transfected with control, NTCP, or host gene targeting siRNAs from the Silencer Select Human Druggable Genome siRNA Library V4 (Thermo Fisher Scientific, 4397922); host genes siRNA plates were screened as described in the *Material* and *Methods* section. After 2 days of transfection, the cells were inoculated with the HBV/NL reporter virus. At 8 dpi, the luciferase assays were performed, and the NL activity was measured and presented as a percentage relative to control siRNA-transfected cells. Of the 2,200 host genes, only 14 genes showed an average of ≥70% reduction of the NL activity upon silencing with a minimum of two independent siRNAs. **(C)** The KIF4 mRNA levels in the liver tissues of patients with chronic HBV infection (n = 122) and healthy subjects (n = 6) (GEO accession number GSE83148). **(D)** HepG2-hNTCP cells were transfected with si-control, si-NTCP, or siRNAs against KIF4 (si-1, and si-2) for 2 days and then inoculated with the HBV/NL reporter virus. At 8 dpi, the cells were lysed, and the luciferase assays were performed, and the NL activity was measured, normalized to cell viability, and plotted as fold changes, relative to control siRNA-transfected cells. **(E)** HepG2-hNTCP cells were transfected with control siRNA or siRNAs against KIF4 (si-1 and si-2); the total protein was extracted after 3 days. The expression of endogenous KIF4 (*upper panel*) and β-actin (loading control) (*lower panel*) was analyzed by immunoblotting with the respective antibodies. Statistical significance was determined using Student's *t*-test (***, $P < 0.001$). For panel (C), statistical significance was evaluated by GEO2R.

## KIF4 regulates HBV and HDV entry into host cells

Then, using particular siRNA, we suppressed KIF4 expression and examined the stage of the HBV life cycle that is controlled by KIF4. NTCP plays a role in the specific binding of HBV to the host cell surface by interacting with the preS1 region of HBV's large surface protein (LHB) [3]. We investigated the attachment of a fluorescence-labeled preS1 peptide (6-carboxytetra-methylrhodamine-labeled preS1 peptide, or TAMRA-preS1) to HepG2-hNTCP (Fig 3A). The

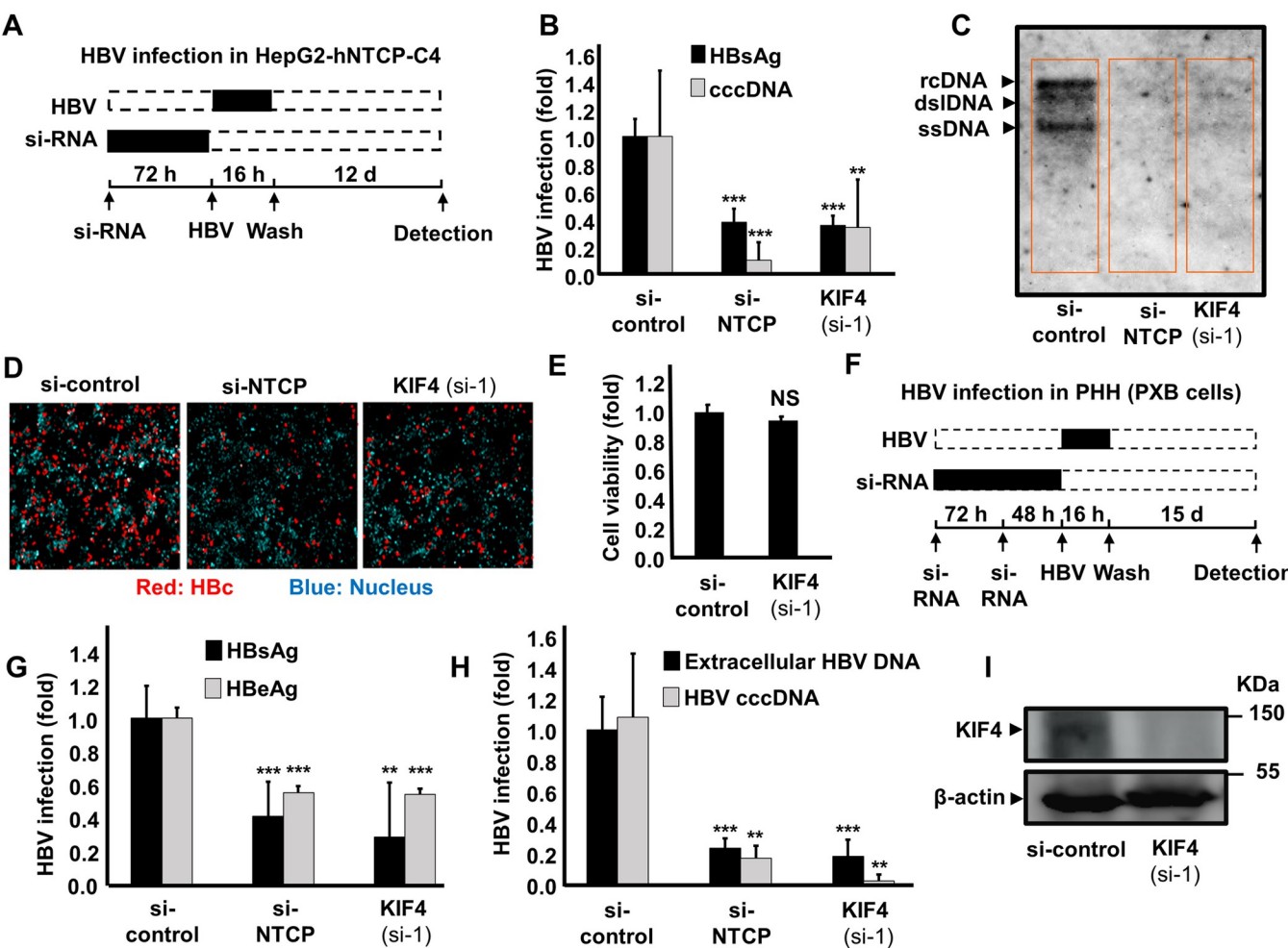

**Fig 2. Decreased KIF4 expression suppressed HBV infection in HepG2-hNTCP and primary human hepatocytes (PHH). (A)** Schematic diagram depicting the scheme for siRNA transfection and subsequent HBV infection in HepG2-hNTCP; HepG2-hNTCP cells were transfected with si-control, si-NTCP, or si-KIF4 (si-1) for 72 h and then inoculated with HBV at 6,000 GEq/cell in the presence of 4% PEG8000 for 16 h. After free HBV were washed out, the cells were cultured for an additional 12 days, followed by the detection of different HBV markers. Black and dashed boxes indicate the interval for treatment and without treatment, respectively. **(B)** HBsAg secreted into the culture supernatant was collected at 10 dpi, quantified by ELISA, and presented as fold changes, relative to the values of control siRNA-transfected cells. **(B)** HBV cccDNA, **(C)** Intracellular HBV DNA, and **(D)** HBc protein in the cells were detected at 13 dpi by real-time PCR, Southern blot analysis, and immunofluorescence, respectively. Red and blue signals in panel **(D)** depict the staining of HBc protein and nucleus, respectively. **(E)** Cell viability was also examined by the XTT assay. **(F)** Schematic diagram showing the scheme for siRNA transfection and the subsequent HBV infection in primary human hepatocytes (PXB); primary human hepatocytes were twice transfected with si-control, si-NTCP, or si-KIF4 (si-1) for consecutive 72 h and 48 h, followed by HBV inoculation at 1,000 GEq/cell in the presence of 4% PEG8000 for 16 h. After being washed, the cells were cultured for an additional 15 days. **(G)** HBsAg and HBeAg secreted into the culture supernatant were quantified by ELISA and Chemiluminescent Immuno-Assay, respectively, and the data were presented as fold changes, relative to the values of control siRNA-transfected cells. **(H)** Extracellular HBV DNA in the culture supernatant and HBV cccDNA (extracted at 8 dpi) were quantified by real-time PCR. In all infection assays including siRNA transfection, control siRNA and NTCP-targeting siRNA were used as negative and positive controls, respectively. **(I)** Primary human hepatocytes were twice transfected with siRNAs (as shown in Fig 2F); the total protein was extracted and KIF4 (*upper panel*) and β-actin (loading control) (*lower panel*) expression were analyzed by immunoblotting with the respective antibodies. All assays were performed in triplicate and included three independent experiments. Standard deviations are also shown as error bars. For HBV cccDNA in panel **(H)**, the assay was performed in triplicate, and data from two independent experiments were pooled (n = 6). Statistical significance was determined using Student's *t*-test (**, $P < 0.01$; ***, $P < 0.001$; NS, not significant).

preS1 binding assay was conducted with si-NTCP as a positive control to validate the specificity of the observed TAMRA-preS1 signals. KIF4 silencing dramatically reduced the interaction between TAMRA-labeled preS1 and NTCP as identified by IF (Fig 3A, left pictures) and shown by signal intensities (Fig 3A, right panel). These findings support the notion that KIF4 is essential for the interaction of HBV and cell surface NTCP. Using a luciferase reporter

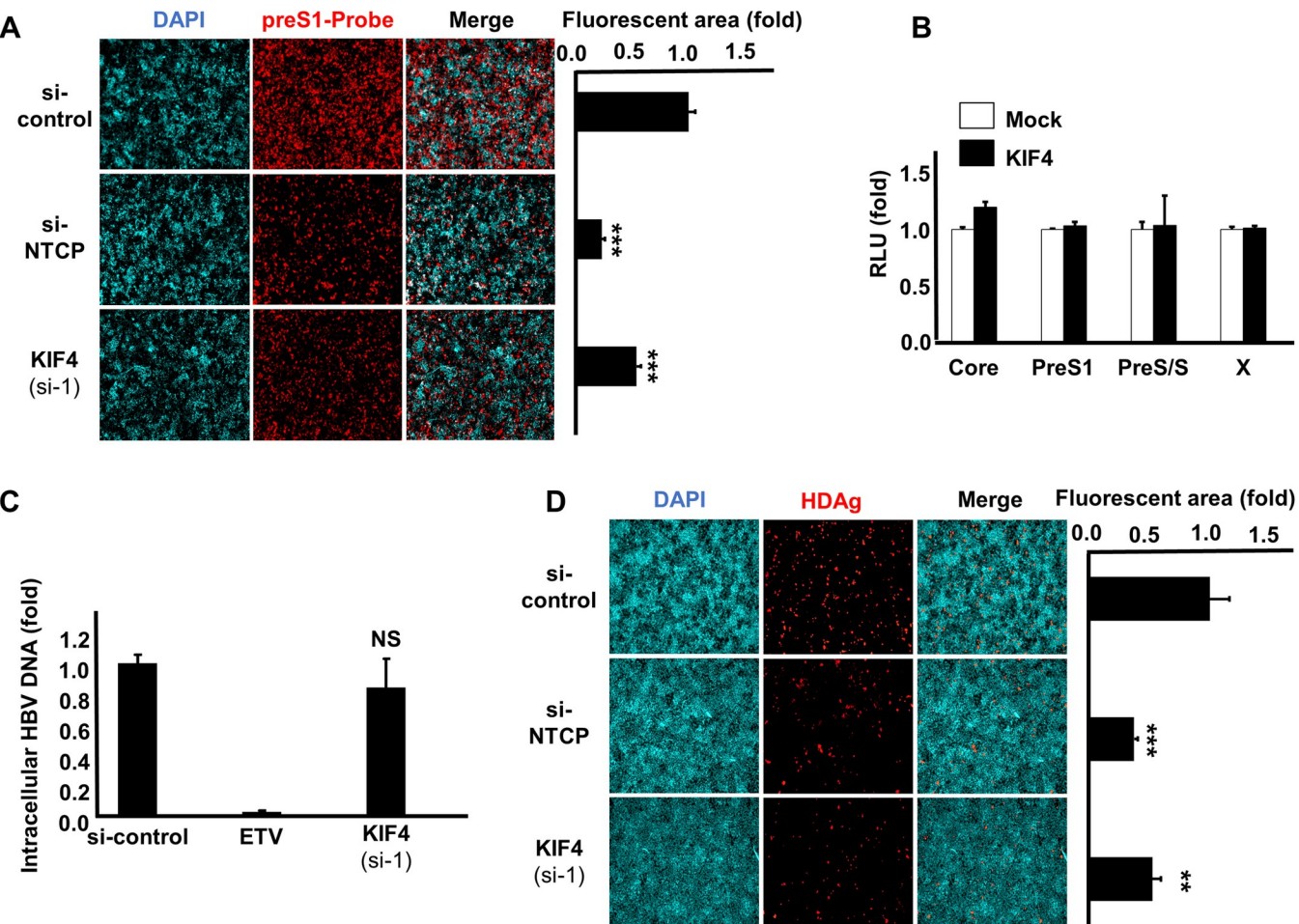

**Fig 3. KIF4 knockdown blocks HBV entry and HDV infection into host cells.** **(A)** HepG2-hNTCP were transfected with si-control, si-NTCP, or si-KIF4 (si-1) for 72 h and then incubated with 40 nM C-terminally TAMRA-labeled and N-terminally myristoylated preS1 peptide (preS1 probe) for 30 min at 37˚C (*left panel*); Red and blue signals indicate the preS1 probe and the nucleus, respectively. The relative fluorescence intensities are shown in the graph (*right panel*). **(B)** HepG2 cells were transfected with a KIF4 expression vector or empty vector (control) together with plasmid vector carrying HBV promoters (Core, X, preS1, or preS2/S) upstream of the *Firefly* luciferase gene and the pRL-TK control plasmid encoding *Renilla* luciferase. At 2 days post-transfection, the cells were lysed and the dual-luciferase activities were measured; the *Firefly* luciferase values were normalized to those of *Renilla* luciferase readings, and the resulting relative luminescence units obtained from KIF4 transfected cells were presented as fold changes compared to the levels detected in the control transfected cells. **(C)** HepAD38.7-Tet cells were transfected with si-control or si-KIF4 (si-1) or treated with 10 μM entecavir as a positive control in the absence of tetracycline to induce HBV replication; At 4 days post-transfection, the cells were lysed and the intracellular HBV DNA was extracted and quantified by real-time PCR. **(D)** HepG2-hNTCP were transfected with siRNAs (as indicated in **Fig 3A**), and then inoculated with HDV virions at 50 GEq/cell in the presence of 5% PEG8000 for 16 h; the cells were then washed out to remove the free virus particles and cultured for an additional 6 days, followed by detection of HDAg by IF (*left panel*); Red and blue signals indicate HDAg and nuclear staining, respectively. The fluorescence intensities are shown in the graph (*right panel*). All assays were performed in triplicate and included three independent experiments. The data were pooled to assess the statistical significance. Data are presented as mean ± SD. **, $P < 0.01$; ***, $P < 0.001$; NS, not significant.

system for various HBV promoters, we discovered that KIF4 expression did not influence the transcriptional activity of these promoters (Fig 3B). Furthermore, utilizing HBV replicon cells, HepAD38.7-Tet off, which produce HBV pgRNA following tetracycline withdrawal, we discovered that suppressing KIF4 expression did not influence intracellular HBV DNA levels (see Fig 3C). HDV has the same envelope as HBV and hence utilizes also NTCP as a receptor to enter hepatocytes [2]. Consistent with the results obtained in the HBV infection and pre-S1-binding tests, suppressing KIF4 expression reduced HDV infection in NTCP-expressing cells (Fig 3D, left pictures), and the magnitude of KIF4 siRNA suppression on HDV infection

was displayed in Fig 3D, right panel. These findings show that KIF4 primarily controlled NTCP-mediated HBV and HDV entry into cells while having little or no influence on HBV transcription or replication.

## KIF4 regulates surface NTCP expression

We investigated the influence of KIF4 on total and subcellular (both surface and cytoplasmic) NTCP expression after discovering that it is necessary for HBV entrance via regulating the interaction between HBV preS1 and NTCP. Silencing of KIF4 expression did not affect total cellular NTCP protein levels, as shown in Fig 4A, but IF examination revealed that it disrupted

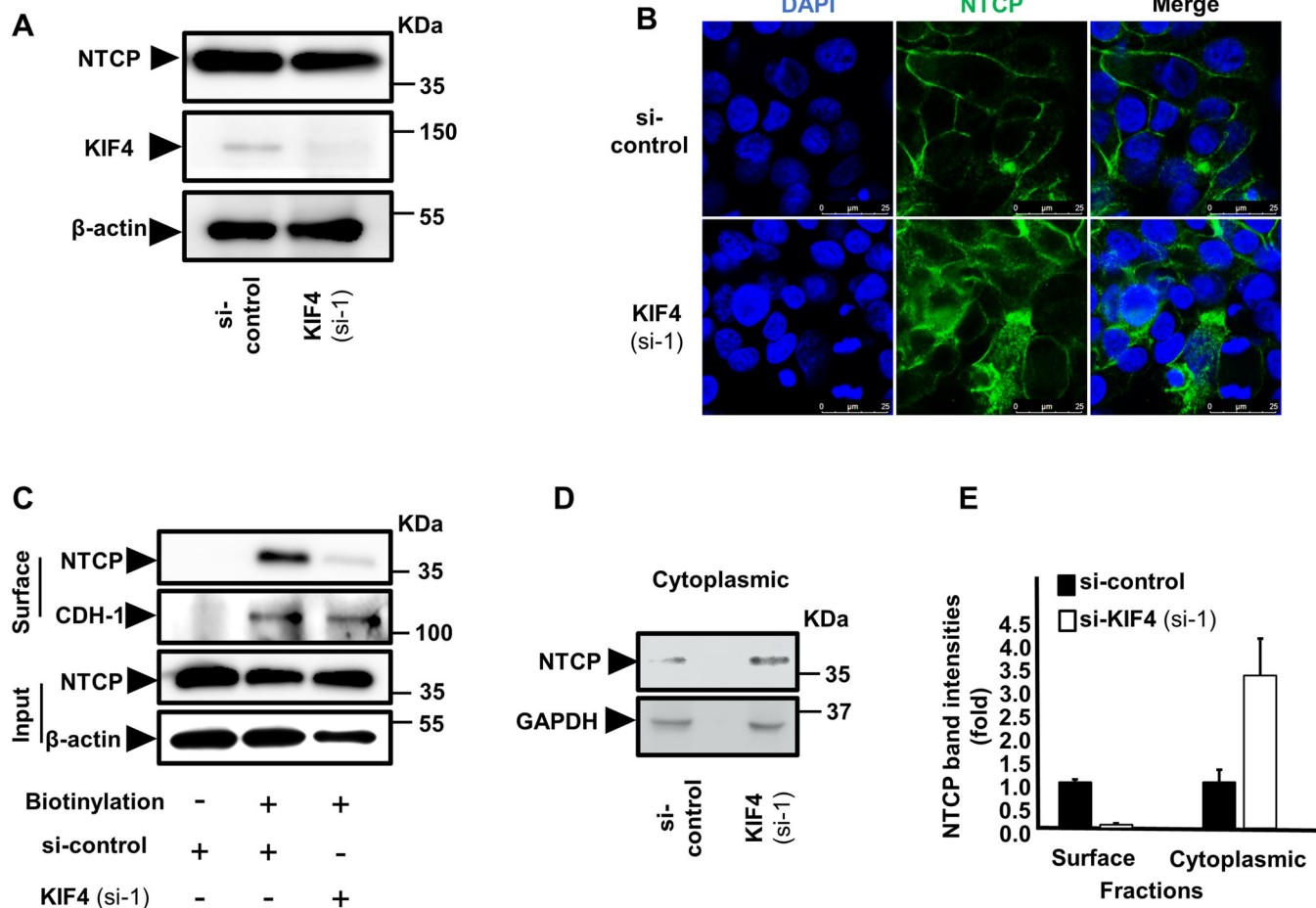

**Fig 4. KIF4 regulates the surface NTCP expression. (A)** HepG2-hNTCP were transfected with either si-control or si-KIF4 (si-1) and incubated for 72 h; the cells were then lysed and the expressions of total NTCP (*upper panel*), KIF4 (*middle panel*), and β-actin (loading control) (*lower panel*) were examined in the whole protein lysate by Western blotting. **(B)** HepG2-hNTCP were transfected with siRNAs (as in **Fig 4A**) and incubated for 72 h; the cells were then fixed with 4% paraformaldehyde, permeabilized with 0.3% Triton X-100, and stained with NTCP antibody and visualized with confocal microscopy. Green and blue signals depict the staining of NTCP (both surface and cytoplasmic), and nuclei, respectively. **(C)** HepG2-hNTCP were transfected with siRNAs (as in **Fig 4A**); at 3 days post-transfection, the cells were surface biotinylated or PBS treated at 4°C for 30 min before cell lysis. After centrifugation and the removal of cell debris, the cell lysates were collected and an aliquot (1/10 volume) was used for the detection of NTCP protein (*Input; upper panel*) and β-actin (loading control) (*Input; lower panels*) by immunoblotting. The remaining cell lysates (9/10 of the original volume) were subjected to pull-down via incubation with pre-washed SA beads for 2 h at 4°C; after washing, the biotinylated surface proteins were eluted and subjected to western blotting in order to detect the surface NTCP and CDH-1 (loading control for surface fraction) (*Surface; upper*, and *lower panels*) with the respective antibodies. **(D)** After transfection with siRNAs (as shown in **Fig 4A**), HepG2-hNTCP cells were lysed and the cytoplasmic fraction was isolated, harvested using the Minute Plasma Membrane Protein Isolation and the Cell Fractionation Kit and then subjected to immunoblotting in order to detect the cytoplasmic NTCP (*upper panel*) and GAPDH (loading control for cytoplasmic fraction) (*lower panel*). **(E)** The intensities of both the surface (normalized to CDH-1) and cytoplasmic (normalized to GAPDH) NTCP bands were quantified by ImageJ software and presented as fold changes relative to the control siRNA-transfected cells.

NTCP surface localization and promoted its accumulation in the cytoplasm (Fig 4B). This conclusion was supported by biochemical investigation, which indicated that silencing KIF4 dramatically decreased surface NTCP in membranous fraction (Fig 4C) while increasing intracellular NTCP protein levels in the cytoplasmic fraction (Fig 4D). Fig 4E depicts band densitometry. It is worth mentioning that silencing KIF4 did not influence cell surface cadherin or cytoplasmic GAPDH protein levels, indicating that KIF4 has a particular effect on NTCP cell surface localization.

## KIF4 motor activity is required for surface NTCP expression

Kinesins, such as KIF4, are motor proteins that hydrolyze ATP to transport different molecules along microtubules [7]. Because KIF4 is necessary for surface NTCP expression, we hypothesized that it may function as a transporter that delivers NTCP to the cell surface. As a result, we investigated the function of KIF4 ATPase in NTCP surface expression. The sequence of an ATPase-null KIF4 was previously described [18] (S1A Fig). To mute endogenous KIF4 expression, we utilized a tailored siRNA sequence (si-KIF4 3′ UTR) that targeted the 3′ UTR region of the KIF4 transcript [19] and compensated for this suppression by transfection with plasmids expressing the Myc-tagged WT or ATPase-null KIF4 sequences. Because they lack the 3′ UTR of endogenous KIF4 transcripts, the mRNAs of these constructs are resistant to si-KIF4 3′ UTR. Cellular fractionation revealed that the expression of WT or ATPase-null mutants did not affect surface cadherin (CHD-1) expression, as predicted. Surface NTCP levels were considerably enhanced when KIF4 depletion was compensated with WT, but not ATPase-null KIF4 protein (Fig 5A). These findings were supported by IF analysis, which revealed significantly greater levels of surface NTCP (Fig 5B), higher binding of HBV preS1 to HepG2-hNTCP cells (Fig 5C), and higher HBV infectivity, as detected by HBc levels (Fig 5D), when endogenous KIF4 silencing was compensated for by WT-KIF4 but not by ATPase-null KIF4. The si-KIF4 3′ UTR exhibited a substantial reduction of KIF4 by real-time RT-PCR and no cellular cytotoxicity, as determined by the XTT assay (S1B and S1C Fig). Overall, our findings indicate that KIF4 ATPase (motor) activity is necessary for NTCP surface expression (transport), and in turn, for NTCP-mediated HBV binding and infection.

## Interaction between KIF4 and NTCP

We hypothesized that an interaction between KIF4 and NTCP across the microtubules is necessary for KIF4 to transport NTCP to the cell surface. We transfected HepG2-hNTCP cells with Halo-tagged KIF4 and examined their intracellular colocalization. Interestingly, IF analysis revealed a significant colocalization between KIF4, NTCP, and α-tubulin (a microtubule marker) (Fig 6A upper panels). Two distinct cross-sectional lines were constructed (Fig 6A, center panels), and the colocalization signal intensities were also displayed along these regions of interest (Fig 6A lower panel). Overlap of KIF4, NTCP, and α-tubulin signal peaks indicated KIF4 and NTCP colocalization along microtubules. The co-immunoprecipitation (co-IP) study confirmed the interaction between NTCP and microtubule-associated KIF4. Myc-tagged KIF4 and HA-tagged NTCP expressing vectors were co-transfected into HEK293-FT cells. We only observed NTCP and the microtubule protein, α-tubulin co-IP when we used Myc antibody to pull down Myc-tagged KIF4, but not when we used control-IgG (Fig 6B). To further investigate the interaction between these two proteins, we prepared recombinant KIF4 and NTCP proteins; after co-incubation of these proteins, we performed a co-IP experiment by pulling down the recombinant Myc-tagged KIF4. Despite the successful pulling down of KIF4, we could not detect the co-precipitation of the recombinant NTCP protein (Fig 6C). Overall, our findings suggest that KIF4 may bind indirectly to NTCP through a complex formation

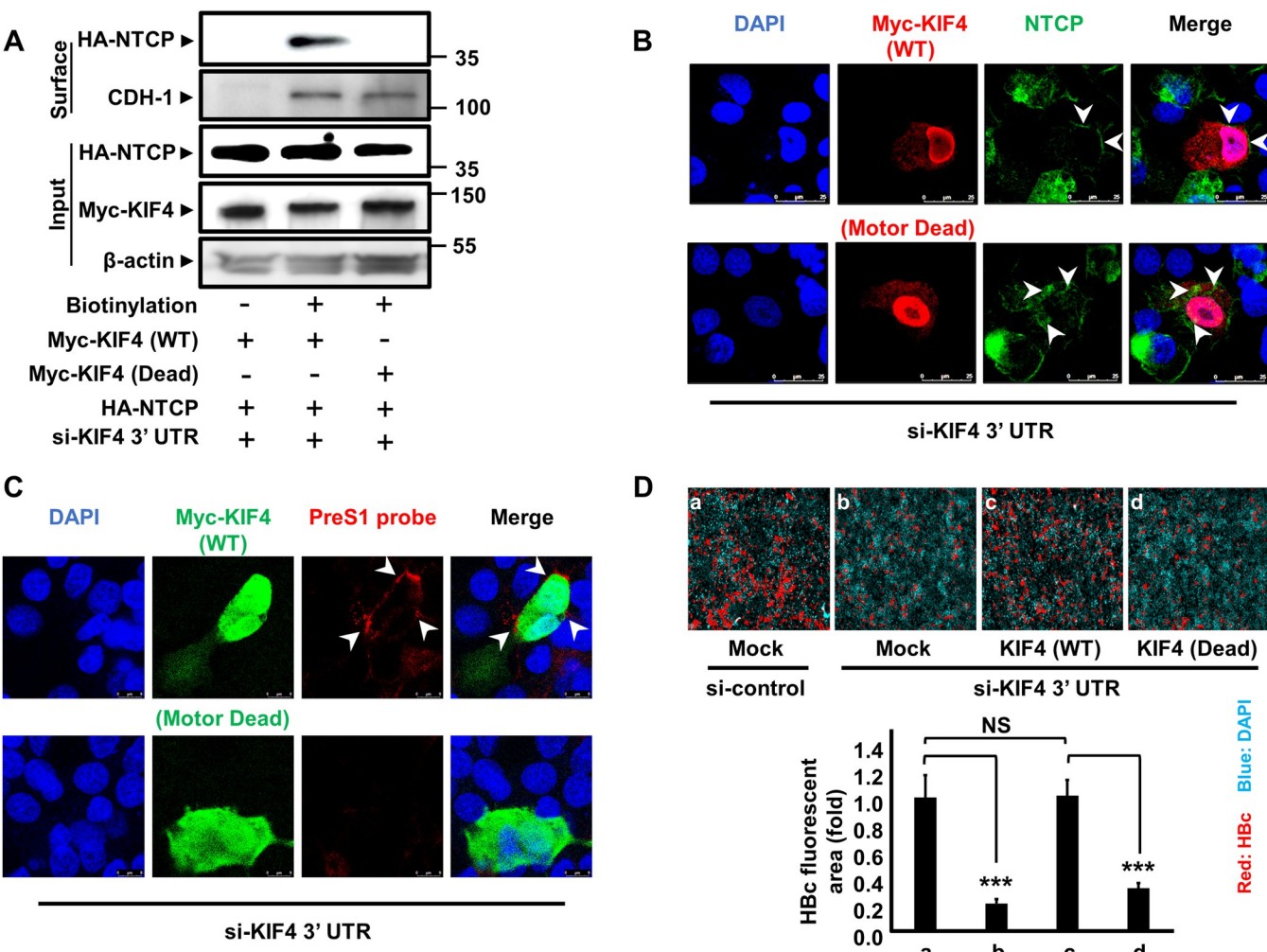

**Fig 5. KIF4 motor activity is required for the surface NTCP expression. (A)** HepG2 cells were transfected with si-KIF4 3′ UTR (targeting endogenous KIF4 mRNA 3'UTR region) along with HA-NTCP, and Myc-KIF4 (WT or ATPase-null) plasmid vectors for 72 h; the cells were then surface biotinylated or PBS treated at 4˚C for 30 min before cell lysis. After centrifugation and the removal of cell debris, cell lysates were collected and an aliquot (1/10 volume) was used for the detection of HA-NTCP (*Input; upper panel*), Myc-KIF4 (*Input; middle panel*), and β-actin (loading control) (*Input; lower panel*) by immunoblotting. The remaining cell lysates (9/10 of the original volume) were subjected to pull-down via incubation with prewashed SA beads for 2 h at 4˚C; after being washed, the biotinylated surface proteins were eluted and subjected to western blotting in order to detect surface HA-NTCP (*Surface; upper panel*) and CDH-1 (loading control for surface fraction) (*Surface; lower panel*) with the respective antibodies. **(B)** HepG2-hNTCP were cotransfected with si-KIF4 3′ UTR and Myc-KIF4 WT (*upper panel*) or motor dead mutant (*lower panel*) plasmid vectors; at 3 days post-transfection, the cells were fixed, permeabilized, stained with the indicated antibodies, and visualized by confocal microscopy. Green, red, and blue signals represent the staining of NTCP, Myc-KIF4, and nuclei, respectively. The arrow heads show NTCP localization in the WT or ATPase-null KIF4 transfected cells. **(C)** HepG2-hNTCP were co-transfected with si-KIF4 3′ UTR and Myc-KIF4 WT (*upper panel*) or motor dead mutant (*lower panel*) plasmid vectors; at 3 days post-transfection, the cells were incubated with TAMRA-labeled preS1 peptide (preS1 probe) for 30 min at 37˚C; the cells were then washed of free preS1 probe, fixed, permeabilized, stained with the indicated antibodies, and visualized by confocal microscopy. Red, green, and blue signals represent preS1 probe, Myc-KIF4, and nuclei, respectively. The arrow heads show preS1 surface localization. **(D)** HepG2-hNTCP were co-transfected with either si-control or si-KIF4 3′ UTR, and Myc-KIF4 (WT or ATPase-null) expression vectors or empty vector for 72 h; the cells were then inoculated with HBV at 6,000 GEq/cell in the presence of 4% PEG8000 for 16 h. After washing free HBV particles, the cells were cultured for an additional 12 days, followed by the examination of HBc protein (*upper panel*) in the cells by IF. Red and blue signals show the staining of HBc protein and nucleus, respectively and HBc fluorescence intensities are shown in the graph (*lower panel*). All assays were performed in triplicate and data from three independent experiments were included. For panel **(D)**, the assay was performed in triplicate and data from two independent experiments were pooled (n = 6). The data were pooled to assess the statistical significance. Data are presented as mean ± SD. ***, $P < 0.001$; NS, not significant.

across the microtubules in the cytoplasm using its ATPase motor domain to transport NTCP to the cell surface. In fact, KIF4 was previously reported to regulate the surface expression of the L1 adhesion molecule through its participation in the antiretrograde transport of vesicles containing L1 to the axon shaft of neuronal cells on the microtubules [20].

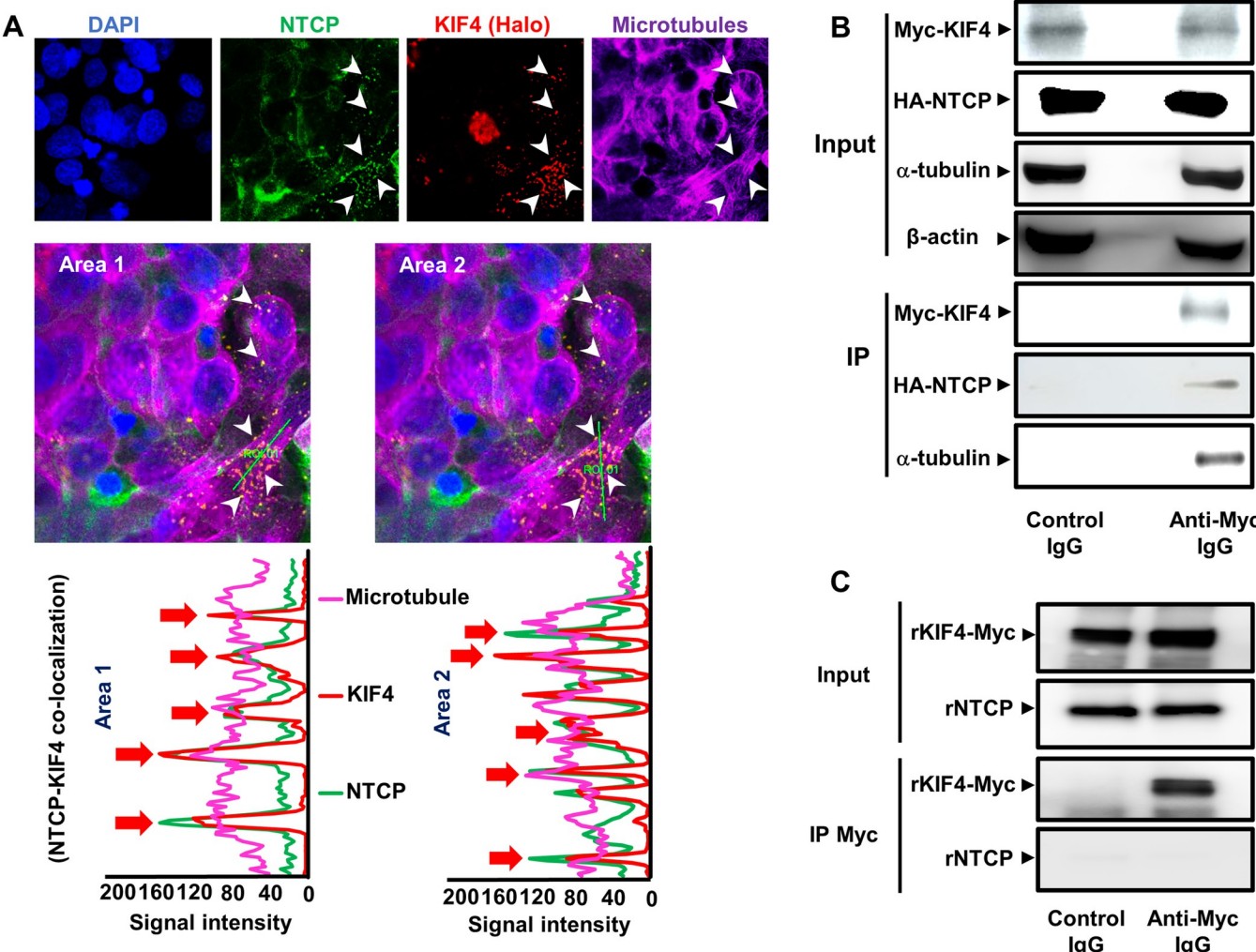

**Fig 6. Interaction of KIF4 and NTCP over microtubule filaments. (A)** HepG2-hNTCP were transfected with a Halo-tagged KIF4 expression vector or empty vector (control). At 48 h post-transfection, the cells were incubated with HaloTag TMR ligand for 15 min at 37˚C; after being washed, cells were fixed, and permeabilized. The cells were stained with antibodies against NTCP and α-tubulin, as indicated in the *Materials* and *Methods* section and examined by confocal microscopy. Blue, green, red, and purple signals indicate nuclear, NTCP, KIF4, and microtubular (α-tubulin) staining, respectively (*upper panel*). White arrows indicate colocalization signals of NTCP and KIF4 over microtubule filaments. The middle panel shows two irrelevant lines crossing different regions of interest within the overlay pattern and represented by their corresponding colocalization signal intensity charts (*lower panel*). **(B)** HEK 293-FT cells were co-transfected with HA-NTCP, and Myc-KIF4 plasmid vectors (at a ratio of 1:1). At 3 days post-transfection, the cells were lysed and an aliquot of the cell lysate (1/10 volume) was used for the detection of Myc-KIF4 (*Input; uppermost panel*), HA-NTCP (*Input; upper middle panel*), α-tubulin (*Input; lower middle panel*), and β-actin (loading control) (*Input; lowermost panel*) by immunoblotting. The remaining cell lysates (9/10 of the original volume) were subjected to co-IP using either isotype control antibody or anti-Myc IgG to pull down Myc-KIF4. Following IP, each sample was analyzed by immunoblotting for Myc-KIF4 (*IP; upper panel*) and both co-immunoprecipitated HA-NTCP (*IP; middle panel*) and α-tubulin (*IP; lower panel*). **(C)** Recombinant NTCP and KIF4 proteins were co-incubated in PBS (in rotation) at 4˚C for 8 hours; an aliquot of the mixture (1/10 volume) was used for the detection of Myc-KIF4 (*Input*; *upper panel*), NTCP (*Input*; *lower panel*) by immunoblotting. The remaining protein mixture (9/10 of the original volume) was subjected to co-IP using either isotype control antibody or anti-Myc IgG to pull down Myc-KIF4. Following IP, each sample was analyzed by immunoblotting for Myc-KIF4 (*IP*; *upper panel*) and NTCP (*IP*; *lower panel*). All assays were performed in triplicate and data from three independent experiments were included. For panel **(C)**, the assay was performed in duplicate and data from three independent experiments were included.

## RXR agonists down-regulate KIF4 expression and block HBV entry by FOXM1 -mediated suppression

The transcription factor Forkhead box M1 (FOXM1) has been shown to increase KIF4A expression [14] and is thus anticipated to affect surface NTCP expression and HBV entry into

hepatocytes. Furthermore, retinoids including retinoid acid receptor (RAR) and retinoid X receptor (RXR) agonists were reported to decrease FOXM1 expression in human oral squamous cell carcinoma [21] and were anticipated to down-regulate its downstream KIF4 expression and, ultimately, HBV entry into human hepatocytes. To test this theory, we looked at how various retinoids affected the attachment of TAMRA-labeled preS1 peptide to cell surface NTCP in HepG2-hNTCP cells (Fig 7A). The preS1 binding assay was done in the presence of NTCP inhibitor, Myrcludex-B, as a positive control to confirm the specificity of the TAMRA-preS1 signals [22]. Interestingly, we discovered that Alitretinoin, a RAR/RXR agonist with potent RXR activity, and Bexarotene, a pan-RXR agonist, significantly reduced TAMRA-labeled PreS1 binding to NTCP as indicated by IF (Fig 7A); however, pan-RAR, ATRA, and RARα-agonist, Tamibarotene, did not affect the preS1 probe binding. These findings indicate that RXR agonists selectively decreased surface NTCP localization and inhibited HBV/NTCP interaction. We then performed cellular fractionation and found that while treatment with Bexarotene did not affect total NTCP expression (Input, Fig 7B), it effectively suppressed the level of NTCP protein in the cell surface fraction, as detected by immunoblotting (Fig 7B). Signal intensity is shown in S2B Fig. Bexarotene treatment of HepG2-hNTCP cells resulted in a significant reduction of both FOXM1 and KIF4 expression, supporting our hypothesis (Fig 7C).

In support of this hypothesis, overexpression of WT FOXM1, but not the dominant negative mutant (FOXM1ΔC), was found to restore both preS1 binding to surface NTCP (Fig 7D) and KIF4 expression levels (Fig 7E) in cells pretreated with 10 μM Bexarotene. As an expected result of the suppression of preS1/NTCP binding by Bexarotene, we found that the pretreatment of HepG2-hNTCP with 10 μM Bexarotene significantly suppressed HBV/NL infection in HepG2-hNTCP cells ($P < 0.001$) (Fig 7F) without altering cell viability (S2A Fig).

In addition to antagonizing HBV/HDV entry by suppressing surface NTCP expression levels, Bexarotene was also expected to antagonize NTCP-mediated conjugated bile salts uptake into hepatocytes. To clarify this point, we analyzed the effect of Bexarotene on sodium-dependent bile salts uptake in primary human hepatocytes. As expected, Bexarotene significantly suppressed bile salts uptake ($p < 0.05$) (Fig 7G).

## Bexarotene pretreatment significantly suppressed HBV and HDV infections in primary human hepatocytes

Bexarotene pretreatment (Fig 8A) significantly reduced susceptibility of primary human hepatocytes (a more realistic model of HBV infection) to HBV infection as shown by the dose-dependent decrease in secreted HBsAg levels ($P < 0.001$) (Fig 8B); the 50% inhibitory concentration ($IC_{50}$) was estimated to be $1.89 \pm 0.98$ μM. Surprisingly, Bexarotene was not harmful to primary human hepatocyte cultures over a wide range of doses, with a 50% cytotoxic concentration ($CC_{50}$) of more than 50 μM (Fig 8C). The selectivity index of Bexarotene ($CC_{50}/IC_{50}$ ratio) was found to be >26. We then changed the timing of Bexarotene administration in order to cover the different stages of HBV life cycle in primary human hepatocytes (Bexarotene was administered as follows: pre = 3 days before HBV inoculation; co = during inoculation of HBV particles from d0 to d1 pi; post = from d4 to d12 pi; and whole = from 3 days before inoculation to d12 pi) (Fig 8D). The expression of HBc protein by IF was used as a marker of HBV infection. While a modest suppression of HBc detection was found when Bexarotene was added co- or post-inoculation, the main suppressive effect of Bexarotene on the level of HBc protein was found when it was administered in (pre), or (whole) settings (Fig 8E [immunofluorescence] and 8F [fluorescence intensity]). There was no apparent difference in Bexarotene-mediated suppression of HBc protein level when it was administered in (pre) or (whole)

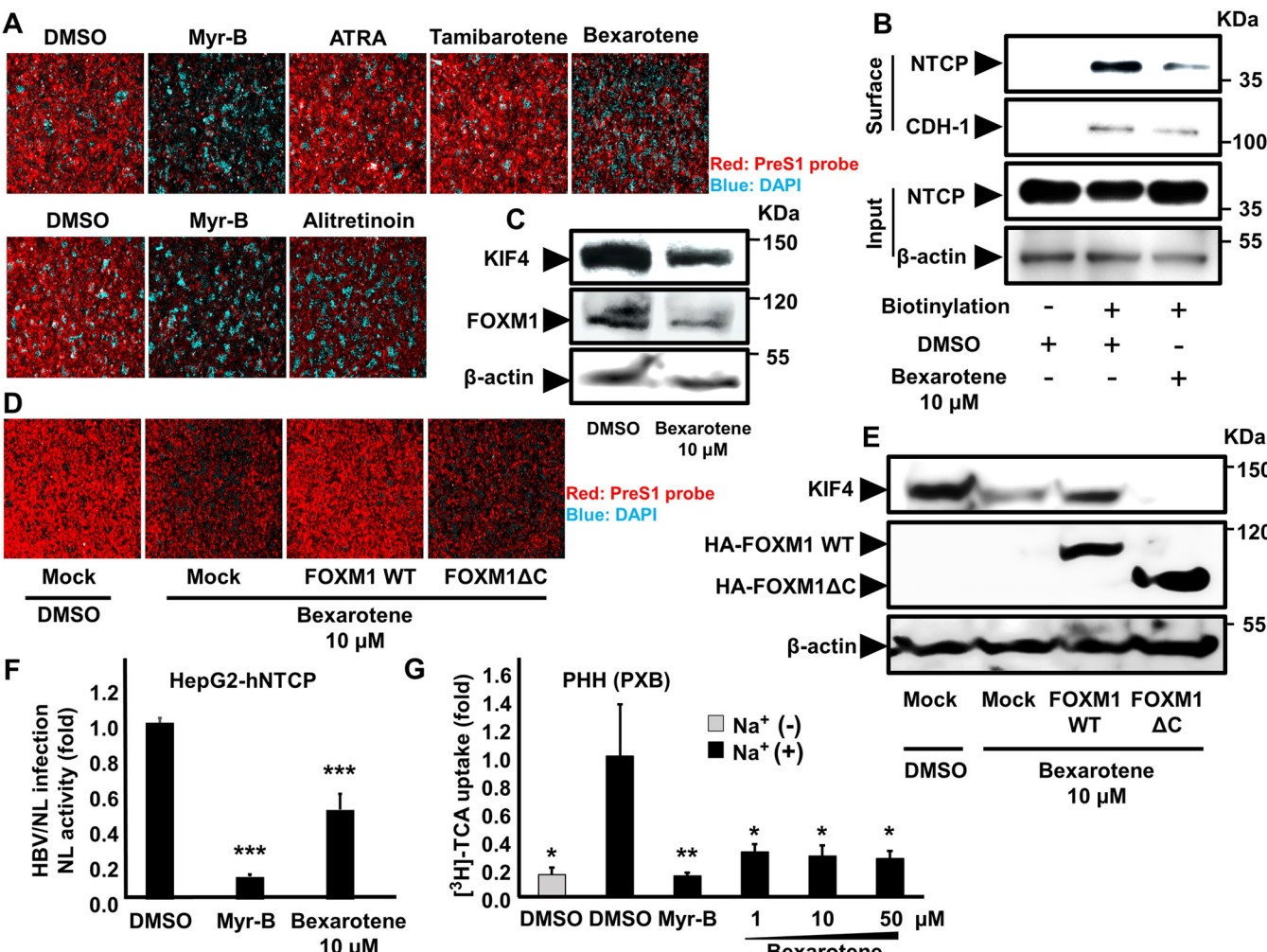

**Fig 7. RXR agonists suppressed KIF4-mediated surface NTCP transport. (A)** HepG2-hNTCP pretreated with DMSO, 10 μM of the indicated compounds (Bexarotene, ATRA, Tamibarotene) (*upper panel*), or Alitretinoin (10 μM) (*lower panel*) for 72 h were incubated with TAMRA-labeled preS1 peptide (preS1 probe) for 30 min at 37°C and then examined by fluorescence microscopy. DMSO-treated cells were incubated with a preS1 probe either in the absence (negative control) or presence (positive control) of 100 nM Myrcludex-B (Myr-B); Red and blue signals indicate preS1 probe and the nucleus, respectively. **(B)** HepG2-hNTCP were treated with DMSO or 10 μM Bexarotene for 72 h, the cells were then surface biotinylated or PBS treated at 4°C for 30 min prior to cell lysis, the cell lysates were collected and an aliquot (1/10 volume) was used for detection of NTCP protein (input) (*Input; upper panel)* and β-actin (loading control) by immunoblotting (*Input; lower panel*). The remaining cell lysates (9/10 of the original volume) were subjected to pull-down via incubation with pre-washed SA beads for 2 h at 4°C; after washing, the biotinylated surface proteins were eluted and subjected to western blotting to detect the surface NTCP (*Surface*; *upper* panel) and CDH-1 (loading control for surface fraction) (*Surface*; *lower panel*) with the respective antibodies. **(C)** HepG2-hNTCP cells treated with DMSO or 10 μM Bexarotene for 72 h were lysed and total cell lysates were subjected to immunoblotting to detect the protein expression levels of KIF4 (*upper panel*), FOXM1 (*middle panel*), and β-actin (loading control) (*lower panel*) with their corresponding antibodies. **(D)** HepG2-hNTCP were transfected with HA-tagged FOXM1 WT, HA-tagged FOXM1ΔC dominant negative mutant vectors, or empty vector. 8 h post-transfection, the cells were treated with DMSO or 10 μM Bexarotene for 72 h and then incubated with TAMRA-labeled preS1 peptide (preS1 probe) for 30 min at 37°C and examined by fluorescence microscopy. Red and blue signals indicate preS1 probe and the nucleus, respectively. **(E)** HepG2-hNTCP were transfected with HA-tagged FOXM1 WT, HA-tagged FOXM1ΔC dominant negative mutant vectors, or empty vector. 8 h post-transfection, the cells were treated with DMSO or 10 μM Bexarotene for 72 h and then lysed and total cell lysates were subjected to immunoblotting to detect the protein expression levels of endogenous KIF4 (*upper panel*), ectopically expressed WT or mutant HA-FOXM1 (*middle panel*), and β-actin (loading control) (*lower panel*) with their corresponding antibodies. **(F)** HepG2-hNTCP were pretreated with DMSO or 10 μM Bexarotene for 72 h; then DMSO and Bexarotene were withdrawn from the culture medium 3 h before HBV/NL inoculation, and the cells were inoculated with the HBV/NL reporter virus for 16 h. DMSO-pretreated cells were concomitantly treated with or without 100 nM Myr-B during HBV/NL inoculation. At 8 dpi, the cells were lysed, luciferase assays were performed, and NL activity was measured, and then plotted as fold changes, relative to the values of control DMSO-pretreated cells. **(G)** Primary human hepatocytes pretreated with DMSO or different concentrations of Bexarotene (1 μM, 10 μM, and 50 μM) for 72 h were incubated with [$^3$H]-taurocholic acid at 37°C for 15 min in a sodium-containing buffer. After washing, the cells were lysed and intracellular radioactivity was measured and plotted as fold changes relative to control DMSO-pretreated cells (*black bars*). Sodium-dependent [$^3$H]-TCA uptake was assessed in DMSO-pretreated cells either in the absence (negative control) or presence (positive control) of 100-nM Myr-B. Sodium-independent [$^3$H]-TCA uptake was also evaluated in DMSO-pretreated cells by incubation with [$^3$H]-TCA in a sodium-free buffer (*gray bar*). All assays were performed in triplicate and data from three independent experiments were included. For panel **(G)**, the assay was performed in quintuplicate (n = 5). The data were pooled to assess the statistical significance. Data are presented as mean ± SD. *, $P < 0.05$; **, $P < 0.01$; ***, $P < 0.001$.

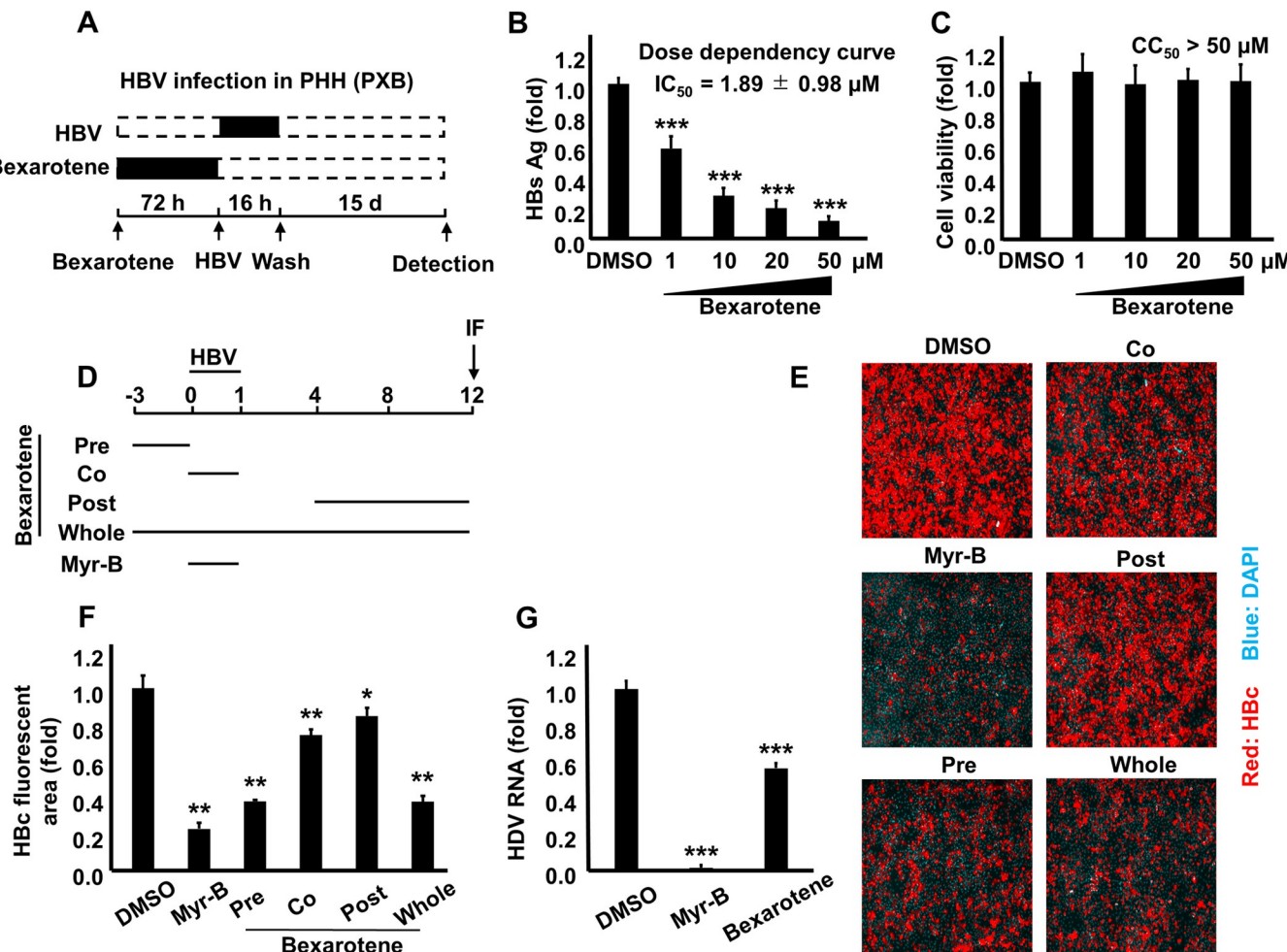

**Fig 8. Bexarotene pretreatment significantly suppressed HBV and HDV infections in primary human hepatocytes (PHH). (A)** A schematic representation showing the protocol used for Bexarotene treatment and subsequent HBV infection in PHH; PHH were pretreated with DMSO or different concentrations of Bexarotene (1 μM, 10 μM, 20 μM, and 50 μM) for 72 h, DMSO and Bexarotene were withdrawn from the culture medium 3 h before HBV infection, and the cells were inoculated with HBV particles at 1,000 GEq/cell in the presence of 4% PEG8000 for 16 h. DMSO-pretreated cells were concomitantly treated with or without 100 nM Myr-B during HBV inoculation. After being washed, the cells were cultured for an additional 15 days. **(B)** HBsAg secreted into the culture supernatant was quantified by ELISA, and the data were presented as fold changes relative to the values of control DMSO-pretreated cells. **(C)** Cell viability were measured by the XTT assay. **(D)** A schematic diagram showing HBV infection protocol; PHH were treated with 15 μM Bexarotene at different time schedule (pre, co, post, and whole) as shown in the Figure. The cells were inoculated with HBV at 1,000 GEq per cell in the presence of 4% PEG8000 for 16 h. Bexarotene non-treated cells were concomitantly incubated with or without 100 nM Myr-B during HBV inoculation. After washing out the free virus particles, the cells were cultured for an additional 11 days. **(E)** HBc protein in the cells was detected by immunofluorescence. Red and blue signals depict the staining of HBc protein and nucleus (dapi), respectively, and HBc fluorescence intensities are shown in panel **(F)**. **(G)** PHH pretreated with DMSO or Bexarotene (50 μM) for 72 h were inoculated with the HDV at 40 GEq/cell in the presence of 5% PEG8000 for 16 h. DMSO-pretreated cells were concomitantly treated with or without 100 nM Myr-B during HDV inoculation. After washing out the free virus particles, the cells were cultured for an additional 6 days and then lysed; RNA was then extracted and HDV RNA was quantified by RT-qPCR. The data are presented as fold differences relative to those of the control DMSO-pretreated cells. All assays were performed in triplicate and data from three independent experiments were included. The data were pooled to assess the statistical significance. For panels **(D and G)**, the assay was performed in triplicate, and data from two independent experiments were pooled. Data are presented as mean ± SD. *, $P < 0.05$; **, $P < 0.01$; ***, $P < 0.001$.

settings. Since the administration of Bexarotene for 3 days before HBV inoculation is the common time frame between (pre) and (whole) settings, our data suggests that Bexarotene mainly exerts its suppression on HBV infection when administered before inoculation. This result is in line with our finding that Bexarotene suppressed surface NTCP localization and subsequent HBV entry; hence, its effect is present mainly when administered before (pre) inoculation. We

investigated the effect of Bexarotene pretreatment on HDV infection. As expected, Bexarotene reduced HDV infection, as shown by a decrease in HDV RNA ($P < 0.001$) (Fig 8G). These findings clearly show that RXR agonists have a suppressive impact on HBV/HDV entry.

## Discussion

The HBV/NL reporter system has previously been described to reflect the early phases of HBV infection [23]. We previously reported the use of this system to screen 2200 druggable human genes and outlined the discovery of MafF and other host factors with anti-HBV function as a result of this screening [16]. The current study describes the proviral host factors that are required for the early phases of HBV infection.

KIF4 belongs to the kinesin superfamily (KIFs). KIFs are ATP-dependent microtubule-based motor proteins that are involved in intracellular transport [11]. The N-terminal motor domain of KIF4 is in charge of ATP hydrolysis and microtubule-binding, whereas the C-terminal domain attaches to cargo molecules such as proteins, lipids, and nucleic acids [24]. KIF4 has previously been implicated in the anterograde transport of cellular proteins such as Integrin beta-1 [25], as well as viral proteins such as retroviral (human immunodeficiency virus [HIV-1], murine leukemia virus, Mason-Pfizer monkey virus, and simian immunodeficiency virus) Gag polyprotein [26] to the cell surface to allow for efficient retroviral particle formation. In line with KIF4's previously described anterograde transport function, we discovered that inhibiting the ATPase motor activity of KIF4 substantially reduced surface and raised cytoplasmic NTCP levels, a phenomenon that was accompanied by a significant suppression of HBV infectivity (Fig 5). These findings demonstrated that KIF4 controlled the anterograde transport of NTCP to the cell surface, influencing its availability as a receptor for HBV/HDV entry at the hepatocyte surface.

One of the selection criteria in our screening for pro- or anti-HBV host genes was that a significant effect on HBV infection should be detected upon silencing the expression of a given gene with a minimum of two independent siRNAs targeting different sequences of its mRNA. Although EGFR was previously reported as an important druggable host-entry cofactor triggering HBV internalization by Iwamoto et al. [27], in our screening, a significant effect was detected with only one EGFR-targeting siRNA sequence; hence, it was excluded from our selection. Because the siRNA concentration required to achieve favorable gene silencing differs from one siRNA sequence to another, and it is difficult to assess the optimum concentration for each si-RNA used in our high throughput screening, we performed the initial screening using siRNA at a 10 nM concentration as recommended by the manufacturer, in contrast to 30–100 nM concentrations used by Iwamoto et al., to silence EGFR expression [27]. Hence, the discrepancy in the results regarding the effect of EGFR silencing on HBV infection could be attributed to the variation in the silencing efficiency of anti-EGFR siRNA used at different concentrations during transfection.

To the best of our knowledge, this is the first study that describes Bexarotene as HBV/ HDV entry inhibitor. Bexarotene has previously been shown to inhibit the early stages of HBV infection when co-inoculated with HBV during the first 24 hours by Song et al. [28]. This impact was influenced in part by RXR-regulated gene expression in arachidonic acid (AA)/eicosanoid biosynthesis pathways, which included the AA synthases phospholipase A2 group IIA (PLA2G2A). However, the specific step of the HBV life cycle (from attachment to cccDNA formation) impacted by Bexarotene was not defined in that study [28]. Furthermore, silencing PLA2G2A expression marginally alleviated Bexarotene's inhibitory impact on HBV infection [28], indicating the presence of other key Bexarotene-dependent mechanisms that are still inhibiting the early stages of HBV infection. In accordance with our data, the effect of

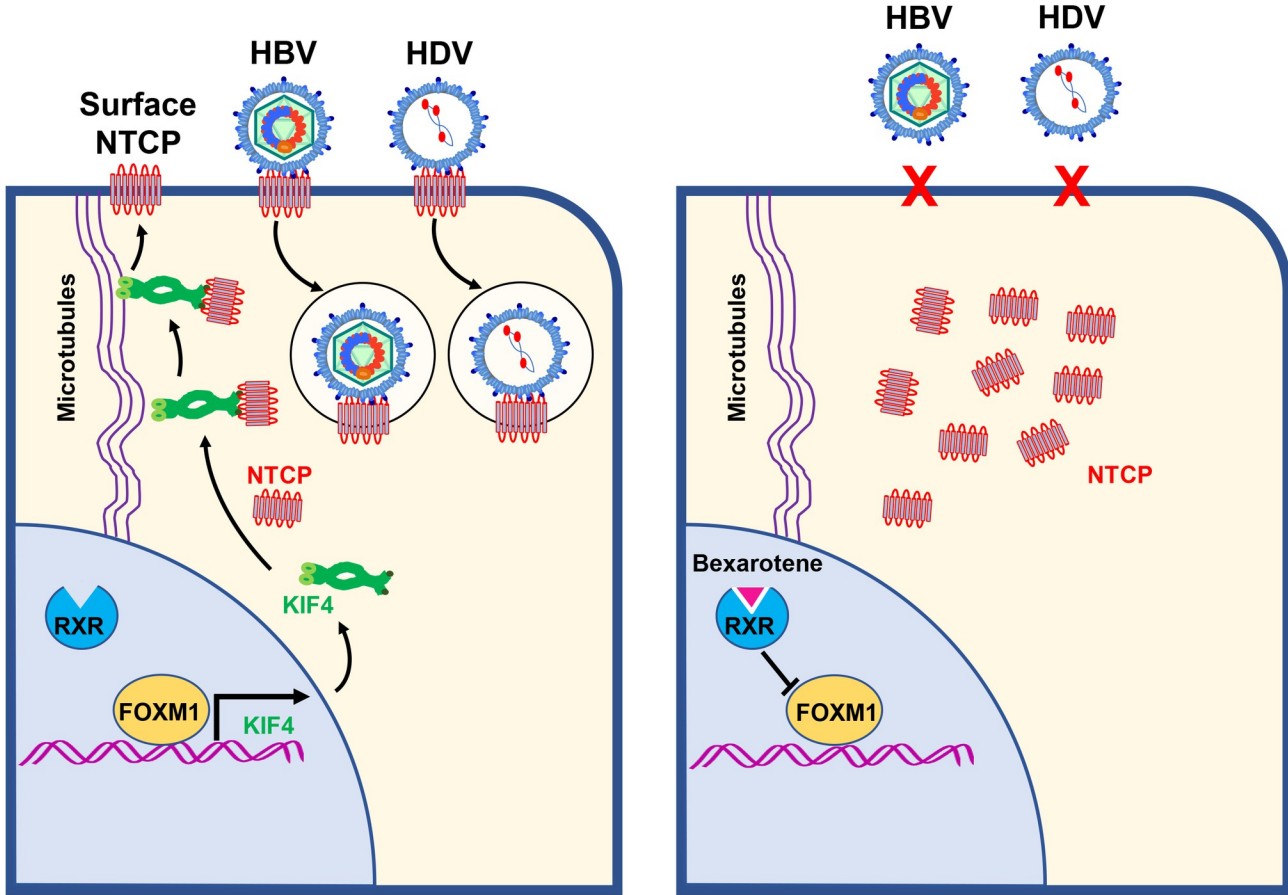

**Fig 9. Proposed model for KIF4 role in NTCP-mediated HBV/HDV entry and impact of KIF4 targeting on HBV/HDV infection.** *Left panel*, KIF4 expression is regulated by transcriptional activator Forkhead box M1 (FOXM1) where cytoplasmic KIF4 pool utilizes microtubules to transport NTCP to the hepatocyte cell surface, where it can act as an entry receptor for HBV and HDV. *Right panel*, pan-RXR agonist Bexarotene downregulates FOXM1 and its downstream KIF4 expression, leading to impaired NTCP surface transport and suppression of HBV/HDV entry into human hepatocytes.

Bexarotene treatment on NTCP expression was also evaluated at the mRNA level in Song *et al.* study [28] and showed no effect on total NTCP expression. Contradicting our findings, Song *et al.* showed that silencing of RXRα did not affect surface NTCP protein levels. However, there are three known subtypes of the RXR family in humans: RXRα, RXRβ, and RXRγ; Bexarotene is a pan-RXR agonist [28,29] and the study by Song *et al.* focused on RXRα based on its abundance in hepatocytes, but this does not exclude the possibility that Bexarotene suppresses NTCP surface localization through its effect on other RXR subtypes. In line with the study by Song *et al.* [28], we found that co-treatment of Bexarotene with HBV inoculation moderately suppressed HBV infection; however, we also showed that the major suppression of HBV infection was obtained when Bexarotene was administrated pre-inoculation (Fig 8E and 8F), suggesting the presence of other significant mechanisms by which Bexarotene exerts its suppressive effect on HBV infection. Furthermore, we found that Bexarotene treatment effectively suppressed cell surface NTCP levels (Fig 7B), and NTCP bile salts transporter activity (Fig 7G). Hence, our data clearly showed that the main mechanism by which Bexarotene suppressed HBV infection is the downregulation of cell surface NTCP levels prior to HBV infection (Fig 9).

E-cadherin was previously reported to facilitate HBV infection and the localization of gly-cosylated NTCP to the cell surface [30]. In that report, a direct interaction between E-cadherin and glycosylated NTCP was found. However, the substantial role of this interaction and the mechanism of E-cadherin regulation of surface distribution of glycosylated NTCP was not investigated [30]. E-cadherin is an essential component of adherens junctions and is required to establish cellular polarity [31]. HBV infects human hepatocytes at the basolateral membrane [32], where NTCP is located exclusively [33]. Hence, E-cadherin may affect HBV entry either through the regulation of hepatocyte polarization and NTCP surface distribution on the baso-lateral membrane, interacting with cell surface NTCP-HBV complex triggering HBV internali-zation into the cell, or through participation in NTCP surface transport. Further analysis is required to explore the function of E-cadherin in HBV entry.

Finally, we identified KIF4 as a critical host factor necessary for effective HBV infection. KIF4 controls the levels of surface NTCP by anterograde transport of NTCP to the cell surface, which is needed for NTCP to function as a receptor for HBV and/or HDV entry. NTCP is the major transporter of conjugated bile salts from the plasma compartment into the hepatocyte. Bexarotene, in addition to suppressing HBV/HDV entry, also suppressed sodium-dependent NTCP-mediated uptake of bile salts into primary hepatocytes (Fig 7G). Although the loss of cell surface NTCP expression in a patient with the homozygous SLC10A1 gene containing a R252H point mutation is associated with higher levels of bile salts in the plasma, there is no evidence of cholestatic jaundice, pruritus, or liver dysfunction. Importantly, the presence of secondary bile salts in the patient's circulation suggested residual enterohepatic cycling of bile salts [34]. Furthermore, in NTCP knockout mice, although a reduced body weight might be observed, most animals showed no signs of cholestasis, inflammation, or hepatocellular dam-age [35].

Myrcludex-B is a lipopeptide and is therefore difficult to administer orally. The availability of other orally administrated drugs may, however, increase drug feasibility and ease of use. Several compounds have to date been reported to suppress NTCP-mediated HBV/HDV entry: Vanitaracin A, irbesartan, ezetimibe, and ritonavir, which directly block NTCP transporter activity, reduce LHBs-dependent viral infection [36–40]. Epigallocatechin-3-gallate (EGCG) and Ro41-5253 decrease cell surface expression of NTCP and reduce HBV infection [41,42]. Bexarotene belongs to this group of compounds reported to suppress NTCP surface expression and showed the highest suppressive effect on HBV infection compared to EGCG and Ro41-5253 (Table 1). Although Bexarotene suppressed both the transcription factor FOXM1, and KIF4, which may result in more side effects irrelevant to HBV/HDV entry inhibition, we did not find any cytotoxic effect of Bexarotene on hepatocytes using concentrations above 50 μM

**Table 1. Chemical compounds targeting NTCP-mediated HBV entry.**

| Name | anti-HBV $IC_{50}$ (μM) | Suppress Bile Salts Uptake | Reference |
|---|---|---|---|
| Compounds blocking NTCP/PreS1 interaction | | | |
| Vanitaracin A | 0.61 ± 0.23 | Yes | [36] |
| Ezetimibe | 1–40 | Yes | [39] |
| Irbesartan | 6.25–12.5 | Yes | [37] |
| Ritonavir | 6.25–12.5 | Yes | [37] |
| (SCY450) Cyclosporine Derivative | 0.9 | No | [65] |
| Compounds affecting total and/or surface NTCP expression | | | |
| Bexarotene | 1.89 | Yes | |
| EGCG | 10–20 | Yes | [42] |
| Ro41-5253 | 5–10 | Yes | [41] |

(CC$_{50}$ > 50 μM) in our in vitro experimental system. Additionally, Bexarotene is approved by the FDA for the treatment of cutaneous lymphoma. Furthermore, Bexarotene IC$_{50}$ for suppressing HBV infection (IC$_{50}$ 1.89 ± 0.98 μM) is much lower than that reported to induce suppression of cancer cells (IC$_{50}$ 50–152 μM) [43,44]. Because HBV entry inhibition reduces the intrahepatic cccDNA pool [45], entry inhibitors are expected to be useful in preventing de novo infection in clinical settings such as vertical transmission and HBV recurrence post-LT. While further in vivo data are still required to assess the efficiency and safety of NTCP-targeting drugs, including Bexarotene, the available data suggest their probable future use as prophylactic treatment for HBV infection. HBIG is used to inhibit HBV vertical transmission [46]. Furthermore, extended therapy with HBIG in conjunction with a nucleos(t)ide analog is necessary following liver transplantation (LT) to reduce the HBV recurrence rate to less than 10% in 1–2 years post-transplantation [47]. HBV DNA integration into the host genome is known to be a risk factor for cancer development [48] and is reported to occur in the early weeks after infection [49,50]. An efficient prophylactic approach using combination of drugs which inhibits HBV cell entry (NTCP surface expression inhibitors, and NTCP-preS1 interaction direct blockers) might be beneficial as an alternative to HBIG to efficiently suppress NTCP-mediated HBV infection and prevent HBV-related complications like HBV DNA integration and HBV-induced liver cancer. These data strongly suggest that the further study of Bexarotene and its functional analogs for the development of a new class of anti-HBV agents is necessary.

## Materials and methods

### Cell culture

HepG2, HepG2-hNTCP-C4, HepAD38.7-Tet, primary human hepatocytes (Phoenixbio; PXB cells), and HEK 293FT cells were cultured as previously described [16]. For maintenance, HepG2-hNTCP cells were cultured in 400 μg/mL G418 [51], while HepAD38.7-Tet cells were cultured in 0.4 μg/mL tetracycline that is withdrawn from the medium upon induction of HBV replication [52].

### Reagents and compounds

Sulfo-NHS-LC-Biotin (A39257) was purchased from Invitrogen. Myrcludex-B was provided by Dr. Stephan Urban, at the University Hospital Heidelberg. Bexarotene (SML0282), ATRA (R2625), Tamibarotene (T3205), Alitretinoin (R4643), Entecavir, and DMSO were all purchased from Sigma-Aldrich.

### Human genome siRNA library screening

siRNA screening was performed as reported previously [53]. Briefly, HBV host factors were screened using the Silencer Select Human Druggable Genome siRNA Library V4 transfection in HepG2-hNTCP cells. siRNAs were arrayed in 96-well plates, and negative control siRNA and si-NTCP were added to control the data obtained from each of the 96-well-plates. siRNAs with different sequences targeting the same genes were distributed across three plates (A, B, and C). Plates utilized in this screening are described elsewhere [16].

### HBV/NL preparation and infection assay

Reporter HBV/NL particles carrying recombinant HBV virus encoding NL gene were collected from the supernatant of HepG2 cells transfected by pUC1.2xHBV/NL plasmid expressing HBV genome (genotype C) in which the core region is substituted with NL-encoding gene, and pUC1.2xHBV-D helper plasmid carrying packaging-deficient HBV genome as

described previously [23,53]. HBV/NL infection was performed 2 days after siRNA transfection. At 8 dpi, the cells were lysed, and the Nano-Luc reading was measured using the Nano-Glo Luciferase Assay System (Promega, N1150), according to the manufacturer's instructions.

## RNA and DNA transfection

The cells were reverse transfected with siRNAs using Lipofectamine RNAiMAX (Invitrogen) according to the manufacturer's guidelines. Forward siRNA transfection in PXB cells was also performed using Lipofectamine RNAiMAX. Transfection with plasmid DNA was performed using the Lipofectamine 3000 (for HepG2 and HepG2-hNTCP) or Lipofectamine 2000 (For 293FT cells), according to the manufacturer's protocol. siRNA/Plasmid DNA cotransfection in HepG2 or HepG2-hNTCP was implemented with Lipofectamine 2000.

## Plasmids and siRNAs

N-terminal HaloTag KIF4 expressing plasmid (pFN21ASDB3041) was purchased from Promega. The N-terminal Myc-tagged KIF4 (both wild-type and ATPase-null motor inactive mutant) cloned in the pIRESpuro3 expression vector was kindly provided by Dr. Toru Hirota at Cancer Institute of the Japanese Foundation for Cancer Research (JFCR). Myc-tagged KIF4 motor inactive mutant was created by substituting 94 aa lysine in the ATP-binding Walker A consensus site to alanine [54]. HA-tagged NTCP was kindly provided by Dr. Hiroyuki Miyoshi at RIKEN BioResource Research Center, Japan [55]. N-terminally HA-tagged FOXM1 WT and FOXM1ΔC [lacking C-terminal transactivation domain, encoding amino acids 1–616, and serving as dominant negative mutant [56]] constructs were created by PCR amplification of the sequence encoding FOXM1 using HaloTag FOXM1 isoform C expressing plasmid (pFN21AB6289, Promega) as the template; the resulting amplification products were digested by *Xho*I and *Xba*I and subcloned into *Xho*I/*Xba*I digested pcDNA3.1 (Invitrogen); the primers used in PCR amplification included forward primer 5′-CGGCCGCTCGAGATGTACCCATA CGATGTTCCAGATTACGCTAAAACTAGCCCCCGTCGGCC-3′ for both constructs and reverse primers 5′-GGGCCCTCTAGACTACTGTAGCTCAGGAATAAACTGGG-3′ and 5′-GGGCCCTCTAGACTAGACAGATTTGCTCGGGGTGGAGG-3′ for HA-FOXM1 WT and HA-FOXM1ΔC, respectively. pUC1.2xHBV/NL and pUC1.2xHBV-D plasmids were kindly supplied by Dr. Kunitada Shimotohno at National Center for Global Health and Medicine, Japan. pSVLD3 plasmid was kindly provided by Dr. John Taylor at the Fox Chase Cancer Center, USA. Silencer Select si-KIF4 (si-1, s24406; si-2, s24408), si-NTCP (s224646), control siRNA (#1), and customized si-KIF4 3′ UTR targeting endogenous KIF4 mRNA 3'-UTR region (5′-GGAAUGAGGUUGUGAUCUUTT-3′) were purchased from Thermo Fisher Scientific.

## HBV infection assay

HBV (genotype D) particles were concentrated from the culture supernatant of HepAD38.7 Tet cells as described elsewhere [51]. HepG2-hNTCP and primary human hepatocytes (PXB) were inoculated with HBV at 6000 and 1000 genome equivalent (GEq)/cell, respectively, as described previously [16].

## HBV preS1 binding assay

HBV preS1 peptide spanning 2–48 amino acids of the preS1 region with N-terminal myristoylation, and C-terminal 6-carboxytetramethylrhodamine (TAMRA) conjugation (preS1 probe)

was synthesized by Scrum, Inc. EZ-Link. The attachment of HBV preS1 peptide to the HepG2-hNTCP cell surface was performed and analyzed as described previously [55].

### HDV infection assay

HDV used in the infection assay was derived from the culture supernatant of Huh7 cells cotransfected with pSVLD3 and pT7HB2.7 as previously reported [57,58]. HepG2-hNTCP and primary human hepatocytes (PXB) were infected with HDV at 40–50 GEq/cell as described previously [59].

### Dual-luciferase reporter assay

HepG2 cells were cotransfected with effector plasmid (Mock or KIF4), the *Firefly* luciferase reporter vectors, and the *Renilla* luciferase plasmid pRL-TK (Promega) as an internal control. The reporter plasmids carrying the entire core promoter (nucleotide [nt] 900–1817), preS1 promoter (nt 2707–2847), preS2/S promoter (nt 2937–3204), or Enh1/X promoter (nt 950–1373) upstream of the *Firefly* luciferase gene, has been reported previously [60]. At 2 days after transfection, the cells were lysed and the luciferase activities were measured using the GloMax 96 Microplate Luminometer (Promega, GMJ96).

### HBV replication assay

In the absence of tetracycline, HepAD38.7-Tet cells were reverse transfected with si-control or si-KIF4 or treated with 10-μM entecavir as a positive control. At 4 days post-transfection, the cells were lysed and the intracellular HBV DNA was extracted and quantified by real-time PCR [52].

### Indirect immunofluorescence assay

Immunofluorescence assay was basically performed as described previously [27]. Primary antibodies used in the study included rabbit anti-HBc (Neomarkers, RB-1413-A), anti-HDAg, anti-NTCP (Sigma, HPA042727), mouse anti-c-Myc (Santa Cruz, sc-40), and anti-α-tubulin (Sigma, T5168). Alexa Flour555-, Alexa Flour488-, or Alexa Flour647-conjugated secondary antibodies (Invitrogen) were utilized together with DAPI to visualize the nucleus. For Halo tag, live cells were treated with cell-permeant Halotag TMR ligand (Promega, G8251) before paraformaldehyde fixation. Microscopic examination of the infected cells or preS1 binding was performed by fluorescence microscopy (KEYENCE, BZ-X710); the observation of the subcellular localization was performed using a high-resolution confocal microscope (Leica, TCS 159 SP8) as described previously [27].

### NTCP transporter assay

NTCP bile transporter activity has been evaluated in primary human hepatocytes as previously described [36]. Briefly, cells pretreated with DMSO or Bexarotene for 72 h were incubated with [$^3$H]-taurocholic acid in either sodium-containing or sodium-free buffer at 37°C for 15 min to allow [$^3$H]-TCA uptake into the cells. After washing to remove free [$^3$H]-TCA, the cells were lysed and intracellular radioactivity was measured using an LSC-6100 liquid scintillation counter.

### Immunoblot assay

Immunoblotting and protein detection were essentially performed as previously described [61]. Protein detection was performed using the following primary antibodies; mouse

monoclonal E-cadherin antibody (Santa Cruz, sc-8426), anti-GAPDH (Abcam, ab9484), anti-Myc (Santa Cruz, sc-40), anti-α-tubulin (Sigma, T5168), anti-β-actin (Sigma-Aldrich, A5441), anti-FOXM1 (Santa Cruz, sc-271746); and rabbit polyclonal anti-KIF4A (Invitrogen, PA5-30492), anti-NTCP (Sigma, HPA042727), anti-HA (Sigma, H6908). For immunoblotting of free or tagged NTCP, the sample was treated with 250-U Peptide-N-Glycosidase F (PNGase F) to digest N-linked oligosaccharides from glycoproteins before loading to SDS-PAGE [27].

## Cell surface biotinylation and extraction of surface proteins

Cell surface biotinylation was performed to separate the surface proteins with streptavidin beads. The cells were washed with PBS and then incubated with 0.5 mg/mL EZ-Link Sulfo-NHS-LC-Biotin for 30 min at 4˚C to biotinylate the cell surface proteins. After quenching with PBS containing 0.1% BSA and washing with PBS thrice to remove free inactive biotin, the cells were lysed in lysis buffer (150 mM NaCl, 50 mM Tris-HCL PH 7.4, 5 mM EDTA, 1% NP40) containing 1x protease inhibitor (Roche) for 15 min at 4˚C. The cell lysate was centrifuged, and the supernatant was harvested and added to prewashed streptavidin agarose (SA) beads and incubated for 2 h at 4˚C (Pull-down step). Finally, the SA beads were washed with lysis buffer, and the adsorbed proteins were eluted in the sample buffer and subjected to immuno-blot assay as described earlier. The surface adhesion protein E-cadherin (CDH-1) was used as a loading control for biotinylated surface fraction, as reported elsewhere [62].

## Purification of cytoplasmic fraction

The cells were washed with cold PBS, lysed, and subjected to cell fractionation; the cytosolic fraction was isolated from the whole cell lysate using the Minute Plasma Membrane Protein Isolation and Cell Fractionation Kit (Invent Biotechnologies) according to the manufacturer's protocol [63].

## Recombinant proteins

Myc-tagged KIF4 recombinant protein was translated from pIRESpuro3 expression vector using the TNT Coupled Reticulocyte Lysate Systems (Promega, L4611) following manufacturer's recommendation; the recombinant NTCP protein preparation was previously reported [36].

## Co-immunoprecipitation assay

293FT cells were transfected with HA-tagged NTCP and Myc-tagged KIF4 expression plasmids at a 1:1 ratio for the assessment of the possible physical interaction between NTCP and KIF4. At 72 h after transfection, the cells were lysed and subjected to immunoprecipitation with the mouse monoclonal anti-Myc (Santa Cruz) antibody or mouse normal IgG as a negative control. Following IP, the samples were subjected to immunoblotting to detect co-IP HA-NTCP and α-tubulin (microtubule marker) in the pull-down fraction. Cell lysis and co-IP were conducted using the Pierce co-IP Kit (Thermo Fisher Scientific, 26149) according to the manufacturer's instructions. Recombinant NTCP and KIF4 proteins were co-incubated in PBS at 4˚C for 8 hours; co-IP were conducted using the Pierce co-IP Kit (Thermo Fisher Scientific, 26149) according to the manufacturer's instructions.

## DNA and RNA extraction

Intracellular HBV DNA and HBV cccDNA were extracted from the cells using the QIAamp Mini Kit (QIAGEN). For HBV cccDNA, DNA extraction was performed without proteinase K

treatment as previously described [53]. Extracellular HBV DNA was recovered from the supernatant using the SideStep Lysis and Stabilization Buffer (Agilent Technologies, 400900), while RNA extraction was performed using the NucleoSpin RNA XS Kit (MACHEREY-NAGEL) according to the manufacturer's protocols.

## Southern blot analysis

Southern blotting was performed to detect intracellular HBV DNAs as described previously [27].

## qPCR and RT-qPCR

Real-time PCR (for the detection of total HBV DNA and HBV cccDNA) and reverse transcription real-time PCR (for the measurement of HDV RNA) were essentially performed as previously described [53,59,64] using the primer-probe sets; 5′-AAGGTAGGAGCTGGAGCAT TCG-3′, 5′-AGGCGGATTTGCTGGCAAAG-3′, 5′-FAM-AGCCCTCAGGCTCAGGGCAT AC-TAMRA-3′ for HBV DNA, 5′-GTGGTTATCCTGCGTTGAT-3′, 5′-GAGCTGAGGCGG TATCT-3′, 5′-FAM-AGTTGGCGAGAAAGTGAAAGCCTGC-TAMRA-3′ for HBV cccDNA, and 5′-GGACCCCTTCAGCGAACA-3′, 5′-CCTAGCATCTCCTCCTATCGCTAT-3′, 5′-FAM-AGGCGCTTCGAGCGGTAGGAGTAAGA-TAMRA-3′ for HDV RNA. qPCR for Intracellular HBV DNA and HBV cccDNA were performed by the $2^{(-\Delta\Delta CT)}$ method using chromosomal GAPDH DNA sequence (via primer-probe set Hs04420697_g1; Applied Biosystems) as an internal normalization control. Isolated RNA was reverse-transcribed using the High-Capacity cDNA Reverse Transcription Kit (Thermo Fisher Scientific), and the relative levels of the KIF4 mRNA were determined using the TaqMan Gene Expression Assay with the primer-probe set Hs00602211_g1 (Applied Biosystems), while the ACTB expression (primer-probe set 748 Hs99999903_m1) was included as an internal control for normalization [53].

## HBsAg and HBeAg quantification

Cell supernatants were harvested and ELISA quantification of the secreted HBs was performed as described previously [64]. The half-maximal inhibitory concentration ($IC_{50}$) value for Bexarotene was calculated as previously reported [55]. HBe antigen was quantified by a Chemiluminescent Immuno-Assay as described previously [46].

## Cell viability assay

Cell viability was evaluated using the Cell Proliferation Kit II (XTT) according to the manufacturer's guidelines [27].

## Database

Transcriptional profiling of patients with chronic HBV (NCBI Gene Expression Omnibus [GEO] accession number GSE83148) was identified in the GEO public database. The expression data for KIF4 were extracted by GEO2R.

## Statistical analysis

Unless mentioned otherwise, the experiments were performed in triplicates, and the means of data from three independent experiments were calculated and presented in mean ± SD. Statistical significance was determined using Two-tailed unpaired Student's $t$-tests (*, $P < 0.05$; **, $P < 0.01$; ***, $P < 0.001$; NS, not significant). For the KIF4 expression level in chronic HBV

(NCBI [GEO] accession number GSE83148), statistical significance was evaluated by GEO2R to calculate the adjusted *P* value.

## Supporting information

**S1 Fig. (A) A schematic diagram illustrating human KIF4 domains and the key regions is presented at the top of the Figure.** Two sequence alignments show ATP-binding Walker A consensus site in the KIF4 motor domain with lysine 94 (wild-type, *upper sequence*) was mutated to alanine (ATPase-null motor dead mutant, *lower sequence*). **(B)** HepG2-hNTCP were transfected with si-control or si-KIF4 3′ UTR for 48 h; the cells were then lysed and the total RNA content was extracted and the KIF4 expression levels were quantified by RT-qPCR and normalized to the expression of ACTB; or **(C)** the cell viability was examined using XTT assay. Data are presented as fold changes relative to those of the control siRNA-transfected cells. All assays were performed in triplicate, and data from three independent experiments were included. The data were pooled to assess the statistical significance. Data are presented as mean ± SD. ***, $P < 0.001$; NS, not significant.
(TIF)

**S2 Fig. (A) HepG2-hNTCP were exposed to DMSO or different concentrations of Bexarotene (1 μM, 10 μM, and 20 μM) for 72 h; cell viability was then evaluated by XTT assay. (B)** The intensities of surface NTCP bands (normalized to CDH-1) shown in Fig 7B were quantified by ImageJ software and presented as fold changes relative to the control DMSO-treated cells. **(C)** The fluorescent intensities of preS1 signal shown in Fig 7D were measured and presented as fold changes relative to mock-transfected DMSO-treated cells. All assays were performed in triplicate and data from three independent experiments were included. The data were pooled to assess the statistical significance. For panel **(C)**, the assay was performed in triplicate, and data from two independent experiments were pooled. Data are presented as mean ± SD. *, $P < 0.05$; ***, $P < 0.001$; NS, not significant.
(TIF)

## Acknowledgments

We gratefully acknowledge Dr. Stephan Urban at University Hospital Heidelberg for providing Myrcludex-B; Dr. Toru Hirota at JFCR, Japan for providing Myc-tagged KIF4 (both wild-type and motor inactive mutant); Dr. John Taylor at the Fox Chase Cancer Center, the USA for providing pSVLD3 plasmid; Dr. Hiroyuki Miyoshi at RIKEN, Japan, for providing HA-tagged NTCP; and Dr. Haruka Kudo at NIID for technical assistance.

## Author Contributions

**Conceptualization:** Sameh A. Gad, Hussein H. Aly.

**Data curation:** Sameh A. Gad, Masaya Sugiyama, Masataka Tsuge, Kosho Wakae.

**Formal analysis:** Sameh A. Gad, Kosho Wakae, Kento Fukano, Mizuki Oshima.

**Funding acquisition:** Kazuaki Chayama, Takaji Wakita, Hussein H. Aly.

**Investigation:** Sameh A. Gad, Masaya Sugiyama, Kento Fukano, Mizuki Oshima, Yingfang Li.

**Methodology:** Sameh A. Gad, Masaya Sugiyama, Masataka Tsuge, Kosho Wakae, Kento Fukano, Camille Sureau, Noriyuki Watanabe, Asako Murayama, Yingfang Li, Ikuo Shoji, Kunitada Shimotohno, Kazuaki Chayama, Masamichi Muramatsu, Takaji Wakita, Tomoyoshi Nozaki, Hussein H. Aly.

**Project administration:** Hussein H. Aly.

**Resources:** Masaya Sugiyama, Masataka Tsuge, Kosho Wakae, Kento Fukano, Camille Sureau, Noriyuki Watanabe, Takanobu Kato, Asako Murayama, Yingfang Li, Ikuo Shoji, Kunitada Shimotohno, Kazuaki Chayama, Masamichi Muramatsu, Takaji Wakita, Tomoyoshi Nozaki, Hussein H. Aly.

**Software:** Noriyuki Watanabe, Takanobu Kato, Yingfang Li.

**Supervision:** Tomoyoshi Nozaki, Hussein H. Aly.

**Validation:** Sameh A. Gad, Hussein H. Aly.

**Writing – original draft:** Sameh A. Gad, Hussein H. Aly.

**Writing – review & editing:** Camille Sureau, Takanobu Kato, Ikuo Shoji, Kunitada Shimotohno, Kazuaki Chayama, Masamichi Muramatsu, Takaji Wakita, Hussein H. Aly.

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
