## [Decision Letter · Decision Letter 0]

31 Oct 2021

Dear Dr. Aly,

Thank you very much for submitting your manuscript "The kinesin KIF4 mediates HBV/HDV entry through regulation of surface NTCP localization and can be targeted by RXR agonists in vitro." for consideration at PLOS Pathogens. As with all papers reviewed by the journal, your manuscript was reviewed by members of the editorial board and by several independent reviewers. The reviewers appreciated the novelty and significance of the work but also raised a number of important issues that should be addressed. In particular, they requested to restore KIF function to KIF KD cells in the context of viral infection, to clarify the mechanisms of bexarotene in relation to what has been reported in the literature vs. in your study, to improve the quality of some of the data (e.g., some western blots), and to use more reliable readout of HBV infection such as HBeAg secretion. Also, as the manipulations used in the study may have pleiotropic effects on viral gene expression and/or antigen secretion, direct measurement of HBV cccDNA as a readout of productive entry (infection) would be helpful in addition to indirect readouts like HBeAg secretion. In light of the reviews (below this email), we would like to invite the resubmission of a significantly-revised version that takes into account the reviewers' comments.

We cannot make any decision about publication until we have seen the revised manuscript and your response to the reviewers' comments. Your revised manuscript is also likely to be sent to reviewers for further evaluation.

Sincerely,

Jianming Hu

Associate Editor

PLOS Pathogens

Jing-hsiung James Ou

Section Editor

PLOS Pathogens

Kasturi Haldar

Editor-in-Chief

PLOS Pathogens

orcid.org/0000-0001-5065-158X

Michael Malim

Editor-in-Chief

PLOS Pathogens

orcid.org/0000-0002-7699-2064

Reviewer's Responses to Questions

**Part I - Summary**

Reviewer #1: The authors used siRNA library screen of HepG2-NTCP cells infected with an HBV reporter to identify 14 genes out of 2,200 genes for which silencing diminished HBV infectivity by > 70%. They focused on KIF4 gene for further characterization because of its proven role for other viruses and its up regulation by HBV (Fig. 1C). They found that KIF4 silencing diminished both HBV and HDV infection, in both HepG2-NTCP cells and primary human hepatocytes. It reduced binding of myristoylated preS1 peptide to cell surface due to diminished NTCP localization on cell surface. Moreover, KIF4 colocalized and physically interacted with NTCP. Its ability to promote NTCP surface localization was abolished by a single amino acid change in its ATP binding site (K94A). As an extension of this, they found bexarotene could reduce KIF4 protein level to diminish NTCP localization on cell surface, thus reducing binding of preS1 peptide and HBV infectivity. Overall, this is a highly novel and quite important finding in the field if it can be independently confirmed.

Certainly, the current findings were all based on knock-down rather than knock-out of KIF4. Hence some of the data are not so impressive (such as HBsAg in Fig. 2B; the high titer in both si-NTCP and si-KIF4 cells could be attributed to residual inoculum when each cell was inoculated with 6000 HBV genomes. HBeAg would be a more reliable infectivity marker in that case). That leaves open the question of whether KIF4 is essential or there are redundant pathways for NTCP cell surface localization. Similarly, since the authors demonstrated that silencing endogenous KIF4 coupled with transfection with wild-type but not K94A mutant could restore NTCP cell surface localization, it would be extremely helpful to perform infection experiments on top of that, thus further proving a role of KIF4 in sustaining HBV infectivity through its motor activity.

Reviewer #2: In this manuscript, Gad et al. have performed functional siRNA screening using HepG2-hNTCP cell-based infection system, aiming to find host factors that can be therapeutically addressed to control HBV infection. The authors identified KIF4, an ATP-dependent microtubule-based motor protein, as a facilitating host factor for HBV/HDV infection. They provided several lines of evidence showing that KIF4 is a critical regulator for cell surface transport of NTCP. They further demonstrated that small molecules downregulating KIF4 expression, e.g. RXR agonist Bexarotene, can suppress HBV infection and be considered as potential antiviral candidates for HBV/HDV infection.

Overall, this manuscript is well written and the in vitro data illustrating a significant role of KIF4 in transporting NTCP to the cell surface for HBV/HDV infection are convincing. This study has made an important contribution to our understanding about how NTCP is transported to the cell surface, however, whether KIF4-targeted therapy using RXR agonists is a practical approach to inhibit HBV/HDV infection in clinical practice need to be further explored. Below are the concerns and suggestions to be addressed

Reviewer #3: Sameh A. Gad et al report in this manuscript the discovery and mechanistic study of kinesin KIF 4 in HBV/HDV infection of hepatocytes. Through a siRNA screen of host cellular genes essential for the early events of HBV infection (from receptor binding to pgRNA production), the authors found that pgRNA reporter was significantly reduced when knockdown of KIF4 expression. The critical role of KIF4 in HBV infection was confirmed in HepG2-NTCP cells and PHHs. Mechanistic studies showed that knockdown of KIF4 significantly (i) reduced the level of preS1 peptide binding, (ii) compromised HDV infection, (iii) reduced NTCP cell surface localization. The authors further showed that the reduced NTCP cell surface expression in KIF4 knockdown cells can be restored by expression of WT, but not ATPase-null KIF4. Finally, the authors showed that treatment of cells with RXR-specific agonist bexarotene inhibited HBV and HDV infection, which also reduced cell surface NTCP expression.

Overall, the data presented strongly support a role of KIF4 in NTCP cell surface trafficking, which is consistent with the well known function of KIF4 in the anterograde transport of cellular and viral proteins to cell surface. Experimentally, several key issues need to be addressed.

**Part II – Major Issues: Key Experiments Required for Acceptance**

Reviewer #1: Some figures of Western blot are of poor quality, such as Fig. 1E, top panel, Fig. 5B, top panel, and Fig. 7B, top panel.

Reviewer #2: Major comments:

1. The authors stated that “RXR agonists (Bexarotene, and Alitretinoin) down-regulated KIF4 expression via FOXM1-mediated suppression” in the abstract (page 2, line 56), however, experimental data supporting this statement is quite limited. They only showed that FOXM1 and KIF4 protein levels were both decreased in 10 uM Bexarotene-treated cells (Fig. 7C). To further confirm the role of FOXM1 in RXR agonist-mediated blocking of HBV/HDV entry, more experiments should be performed, e.g. rescue experiments using exogenously expressed FOXM1 or FOXM1 dominant negative mutant.

2. Co-immunoprecipitation study cannot definitely prove the physical contact of KIF4 and NTCP (page 12, line 223; page 16, line 309). An in vitro pull down assay using purified recombinant KIF4 and NTCP proteins should be performed to rule out the possibility that these two proteins are co-immunoprecipitated through a third partner present in the lysates of cells co-transfected with KIF4- and NTCP-expressing plasmids.

3. Many HBV/HDV entry inhibitors would interfere normal physiological function of NTCP, mainly bile acid transportation. The authors should examine the effect of RXR agonists on bile acid transporter function of NTCP in vitro and in vivo.

4. Since the discovery of NTCP as the high affinity receptor for HBV/HDV, many studies have identified compounds that can inhibit viral entry, including the most well-known one, Myrcludex. From the data of Fig. 7, it’s apparent that the RXR agonist Bexarotene inhibits HBV/HDV infection to a much lesser extent than MyrB (Figs. 7 and 8) at a much higher dose (10 uM Bexarotene vs. 100 nM MyrB) in vitro. Furthermore, unlike MyrB that interacts directly with NTCP to block HBV/HDV interaction, RXR agonist works through downregulating the expression of FOXM1, a transcription activator of KIF4. Therefore, RXR agonist affects not only the expression of FOXM1 and KIF4 but also other downstream genes of RXR, which may result in more side effects irrelevant to HBV/HDV entry inhibition than MyrB. From the clinical point of view, RXR agonists seem not to be good drug candidates for inhibiting HBV/HDV entry. The authors should discuss these concerns and compare its clinical utility with other HBV/HDV entry inhibitors, in treating HBV infection.

5. In the discussion section, the contents of the paragraphs 2, 3 and 4 are largely overlap those in the results section. These contents should be re-organized to a more concise form. Moreover, the proposed role of KIF4 in the current HBV/HDV entry model should be included in the discussion and better be depicted in a diagram.

Reviewer #3: 1. It is important to perform a complementation assay in KIF4 knockdown cell to show that expression of WT, but not the ATPase-dead KIF4, can restore HBV and HDV infection efficiency.

2. It is curious that 10 μM Bexarotene treatment of HepG2-NTCP cells for 72 h does not alter the levels of total cellular KIF4 in the experiment presented in Fig. 7B, but reduced the level of total cellular KIF4 in the experiment shown in Fig. 7C. Explain?

3. Song et al. showed that bexarotene inhibits HBV infection in HepG2-NTCP cells, HepaRG cells, and primary Tupaia hepatocytes, mainly through modulation of cellular lipid metabolism. In this report, the authors hypothesize that bexarotene reduces the level of transcription factor FOXM1, which is essential for KIF4 transcription. Although the data presented in Fig 7C showed that bexarotene treatment did reduce FOXM1, but the role of FOXM1 in KIF4 expression and NTCP cell surface trafficking remain to be experimentally determined in hepatocytes.

**Part III – Minor Issues: Editorial and Data Presentation Modifications**

Reviewer #1: 1. The authors cited and interpreted data from Dr. Wenhui Li’s group (ref. 28). According to that paper, in which Bexarotene was added during but not prior to HBV infection, it did not alter NTCP cell surface localization. What's the explanation for the discordance?

2. Some writing issues. Line 53: please replace “immunofluorescence (IF)” with “IF”, as the abbreviation already occurred at lines 51-52. Line 84: “nucleocapsid infects human hepatocytes” should be changed to “nucleocapsid enters human hepatocytes”. Line 186: “encouraged its accumulation” is better changed to “promoted its accumulation”.

Reviewer #2: Minor comments:

1. Page 4, line 84: Because nucleocapsid cannot infect hepatocytes, the sentence “When HBV nucleocapsid infects human hepatocytes, it is carried to the nucleus” should be better changed to “When HBV infects human hepatocytes, its nucleocapsid is carried to the nucleus”.

2. Page 19, line 352: “pruritis” seems to be a typo for “pruritus”.

3. Fig. 2, only HBsAg and viral DNA data were shown to illustrate the effect of KIF4 knockdown on HBV infection. It’s better to include HBe data in this figure because HBeAg secretion but not HBsAg secretion has been shown to correlate with HBV infection efficacy in the in vitro infection system. (Dr. Urban’s study, Gastroenterology, 2014; 146: 1070-1083)

4. In Fig. 7, the authors applied immunofluorescence staining of alpha-tubulin and co-localization analysis to demonstrate the interaction of KIF4 and NTCP over microtubule filaments. To make the result more solid, detection of microtubule-associated proteins, such as alpha-tubulin, in the KIF4 and NTCP co-immunoprecipitated complex is suggested.

5. EGFR has been also been shown to be a druggable host-entry cofactor (PNAS, 2019; 116: 8487-8492). However, the authors did not find EGFR in their functional screening. This reviewer suggests to include this issue in the Discussion section.

6. Fig. 7B, the signal intensity of each band should be quantitated as in Fig. 4.

Reviewer #3: 1. Reference 16 and 36 are duplicates

PLOS authors have the option to publish the peer review history of their article (what does this mean?). If published, this will include your full peer review and any attached files.

Reviewer #1: No

Reviewer #2: No

Reviewer #3: No
---

## [Decision Letter · Decision Letter 1]

22 Feb 2022

Dear Dr. Aly,

Thank you very much for submitting your manuscript "The kinesin KIF4 mediates HBV/HDV entry through regulation of surface NTCP localization and can be targeted by RXR agonists in vitro." for consideration at PLOS Pathogens. As with all papers reviewed by the journal, your manuscript was reviewed by members of the editorial board and by several independent reviewers. The reviewers appreciated the attention to their concerns in revising the manuscript but two reviewers have some remaining minor concerns. Based on the reviews, we are likely to accept this manuscript for publication, providing that you modify the manuscript according to the review recommendations.

Sincerely,

Jianming Hu

Associate Editor

PLOS Pathogens

Jing-hsiung James Ou

Section Editor

PLOS Pathogens

Kasturi Haldar

Editor-in-Chief

PLOS Pathogens

orcid.org/0000-0001-5065-158X

Michael Malim

Editor-in-Chief

PLOS Pathogens

orcid.org/0000-0002-7699-2064

Reviewer Comments (if any, and for reference):

Reviewer's Responses to Questions

**Part I - Summary**

Reviewer #1: The authors have performed additional experiments and answered critiques from all the three reviewers. The manuscript is much improved.

Reviewer #2: The revisions addressed the reviewer's comments pretty well.

Reviewer #3: The authors addressed my concerns on the previous version of this manuscript with complete satisfaction.

**Part II – Major Issues: Key Experiments Required for Acceptance**

Reviewer #1: (No Response)

Reviewer #2: No.

Reviewer #3: No.

**Part III – Minor Issues: Editorial and Data Presentation Modifications**

Reviewer #1: I accidentally found a paper on the same topic by Hu et al entitled “E-cadherin plays a role in hepatitis B virus entry through affecting glycosylated sodium-taurocholate cotransporting polypeptide distribution” published in Frontiers in Cellular and Infection Microbiology in 2020 (10: 74). In that paper the authors claimed that E-cadherin promotes the cell surface localization of glycosylated NTCP to facilitate HBV infection of HepG2/NTCP cells, HepaRG cells, and PHHs. They found physical interaction between E-cadherin and glycosylated NTCP. Such findings are very similar to the relationship between KIF4 and NTCP. It remains to be seen whether both E-cadherin and KIF4 could modulate HBV infectivity through NTCP localization, or the report on E-cadherin is unreliable. For fairness of literature review this paper on the same topic should be cited and probably discussed briefly in the Discussion section.

Reviewer #2: However, some minor points are suggested to be addressed.

1. I would suggest editing some parts of the “Discussion section” to achieve better logical flow and conciseness. For example, paragraph 3 (lines 340-346) seems to be overlapped a lot with the results and does not add too much insight for the discussion.

2. There are English writing issues in the newly added texts. English editing is suggested.

3. P19, lines 363-364, the author stated that “Bexarotene suppresses NTCP expression ---”. However, this manuscript and DR. Li’s work all showed that Bexarotene does not affect the expression of NTCP at total protein and RNA levels, respectively. Therefore, this sentence should be changed to “Bexarotene suppresses surface NTCP expression (or surface localization) ---”.

4. p31 line 573: 4C � 4oC.

Reviewer #3: No.

PLOS authors have the option to publish the peer review history of their article (what does this mean?). If published, this will include your full peer review and any attached files.

Reviewer #1: No

Reviewer #2: No

Reviewer #3: No

Figure Files:

Data Requirements:

Reproducibility:

References:

---

## [Editor Report · Decision Letter 2]

4 Mar 2022

Dear Dr. Aly,

We are pleased to inform you that your manuscript 'The kinesin KIF4 mediates HBV/HDV entry through the regulation of surface NTCP localization and can be targeted by RXR agonists in vitro.' has been provisionally accepted for publication in PLOS Pathogens.

Best regards,

Jianming Hu

Associate Editor

PLOS Pathogens

Jing-hsiung James Ou

Section Editor

PLOS Pathogens

Kasturi Haldar

Editor-in-Chief

PLOS Pathogens

orcid.org/0000-0001-5065-158X

Michael Malim

Editor-in-Chief

PLOS Pathogens

orcid.org/0000-0002-7699-2064
---

## [Editor Report · Acceptance letter]

17 Mar 2022

Dear Dr. Aly,

We are delighted to inform you that your manuscript, "The kinesin KIF4 mediates HBV/HDV entry through the regulation of surface NTCP localization and can be targeted by RXR agonists in vitro.," has been formally accepted for publication in PLOS Pathogens.

Best regards,

Kasturi Haldar

Editor-in-Chief

PLOS Pathogens

orcid.org/0000-0001-5065-158X

Michael Malim

Editor-in-Chief

PLOS Pathogens

orcid.org/0000-0002-7699-2064